# CDK1 couples proliferation with protein synthesis

Katharina Haneke[1,2], Johanna Schott[1,2], Doris Lindner[1,2], Anne Kruse Hollensen[3], Christian Kroun Damgaard[3], Cyril Mongis[2], Michael Knop[2,4] ⓘ, Wilhelm Palm[5], Alessia Ruggieri[6] ⓘ, and Georg Stoecklin[1,2] ⓘ

**Cell proliferation exerts a high demand on protein synthesis, yet the mechanisms coupling the two processes are not fully understood. A kinase and phosphatase screen for activators of translation, based on the formation of stress granules in human cells, revealed cell cycle–associated kinases as major candidates. CDK1 was identified as a positive regulator of global translation, and cell synchronization experiments showed that this is an extramitotic function of CDK1. Different pathways including eIF2α, 4EBP, and S6K1 signaling contribute to controlling global translation downstream of CDK1. Moreover, Ribo-Seq analysis uncovered that CDK1 exerts a particularly strong effect on the translation of 5′TOP mRNAs, which includes mRNAs encoding ribosomal proteins and several translation factors. This effect requires the 5′TOP mRNA-binding protein LARP1, concurrent to our finding that LARP1 phosphorylation is strongly dependent on CDK1. Thus, CDK1 provides a direct means to couple cell proliferation with biosynthesis of the translation machinery and the rate of protein synthesis.**

## Introduction

Cell growth, proliferation, and progression through the cell cycle strongly depend on the synthesis of new proteins (Pardee, 1989; Polymenis and Aramayo, 2015). On the one hand, cells exert temporal control over the production of specific proteins during the different phases of the cell cycle (Aviner et al., 2013; Stumpf et al., 2013; Tanenbaum et al., 2015). On the other hand, cells also need to adjust the overall rate of protein synthesis to the proliferation rate to maintain cell size and functionality (Foster et al., 2010; Miettinen et al., 2019). It is therefore not surprising that modifications of the translation machinery can affect cell proliferation rates and that deregulation of protein synthesis is increasingly recognized as a major driver of cell transformation (Ruggero and Pandolfi, 2003; Silvera et al., 2010; Truitt and Ruggero, 2016).

A few signaling pathways are known to regulate protein synthesis in response to proliferative cues. The mechanistic target of rapamycin complex 1 (mTORC1), for example, functions as a signaling node that adjusts protein synthesis to cell growth rates and the metabolic status of the cell (Laplante and Sabatini, 2012). mTORC1 directly phosphorylates 4E binding proteins (4EBPs), thereby promoting the translation of a distinct group of mRNAs that strongly depend on the eukaryotic translation initiation factor (eIF) 4E (Gandin et al., 2016; Nandagopal and Roux, 2015). mTORC1 further enhances the translation of mRNAs containing a 5′ terminal oligopyrimidine tract (5′TOP) motif, which includes many mRNAs encoding ribosomal proteins (RPs) and translation factors (Meyuhas and Kahan, 2015).

The protooncogenes Ras and Myc also control protein synthesis to coordinate cellular growth rates with extracellular growth stimuli. While Myc mostly controls translation through transcriptional up-regulation of ribosomal components and translation factors (van Riggelen et al., 2010), the Ras/Erk signaling pathway shares some common downstream signals with mTORC1, including phosphorylation of RPS6 (Roux and Topisirovic, 2018).

While numerous translation factors are known to be phosphorylated (Roux and Topisirovic, 2018), the regulatory impact of phosphorylation is established for only a few factors such as eIF2α, 4EBPs, and eukaryotic translation elongation factor 2 (eEF2; Jackson et al., 2010; Kenney et al., 2014). RPs are also known to carry various posttranslational modifications (Shi and Barna, 2015), yet the role of these modifications in controlling protein synthesis is poorly understood. Recently, a systematic approach to identify translationally relevant phosphorylation sites on RPs revealed that phosphorylation of RPL12 controls the translation of mitosis-specific proteins (Imami et al., 2018).

At the core of the cell cycle, CDKs drive cells through the different phases of the cell cycle. In G1, Cyclin D-CDK4/6 (early) and Cyclin E-CDK2 (late) prepare entry into S phase, where Cyclin A-CDK2 takes over and orchestrates replication, followed by activation of Cyclin A/B-CDK1 promoting passage through G2 and entry into M phase (Malumbres and Barbacid, 2005). Interestingly, CDK1 can substitute for the other CDKs and was found to be

[1]Division of Biochemistry, Mannheim Institute for Innate Immunoscience, Medical Faculty Mannheim, Heidelberg University, Mannheim, Germany;   [2]Center for Molecular Biology of Heidelberg University, DKFZ-ZMBH Alliance, Heidelberg, Germany;   [3]Department of Molecular Biology and Genetics, Aarhus University, Aarhus, Denmark;   [4]Cell Morphogenesis and Signal Transduction, German Cancer Research Center, DKFZ-ZMBH Alliance, Heidelberg, Germany;   [5]Cell Signaling and Metabolism, German Cancer Research Center, DKFZ-ZMBH Alliance, Heidelberg, Germany;   [6]Department of Infectious Diseases, Molecular Virology, Center for Integrative Infectious Diseases Research, University of Heidelberg, Heidelberg, Germany.

Correspondence to Georg Stoecklin: georg.stoecklin@medma.uni-heidelberg.de.

sufficient for driving the mammalian cell cycle (Santamaría et al., 2007). CDK1 has also been linked to the control of protein synthesis during M phase (Shuda et al., 2015; Sivan et al., 2011).

In this study, we made use of the fact that a global decrease in translation initiation is coupled to the assembly of cytoplasmic stress granules (SGs), aggregates that arise through phase separation of stalled mRNAs and associated factors from the surrounding cytosol (Kedersha et al., 2013). To identify novel regulators of protein synthesis, we conducted an siRNA screen against human kinases and phosphatases using SG formation as a visual readout. Since cell cycle–associated kinases were among the primary candidates identified in the screen, we chose to pursue CDK1 and characterize its role in protein synthesis. Our results demonstrate that CDK1 acts outside of mitosis as a general activator of translation that allows direct adaptation of protein synthesis to the rate of cell proliferation.

## Results

### Identification of kinases and phosphatases suppressing SG assembly

With the aim to identify kinases and phosphatases that affect global protein synthesis and/or SG assembly, we knocked down 711 human kinases and 256 phosphatases in HeLa cells stably expressing the SG marker GFP-G3BP1, using four independent siRNAs for each phosphotransferase. After 72 h, cells were fixed and monitored by automated fluorescence microscopy for the presence of SGs. As expected, GFP-G3BP1 was evenly distributed in the cytoplasm in control knockdown (KD) cells (Fig. 1 A). SG formation was detected in a small fraction of the KD cultures, and typically occurred only in a subpopulation of cells (examples in Fig. 1 A). For every phosphotransferase, we calculated a SG score (Repository Table R1, https://doi.org/10.11588/data/EFHOBZ), which reflects both the strength of the phenotype and its reproducibility. The screen identified 54 candidate kinases (8%) and 15 candidate phosphatases (6%) whose KD led to SG formation with a SG score >10 (with ≥2 siRNAs) or >40 (with 1 siRNA; Fig. 1, B and C, Repository Table R2, https://doi.org/10.11588/data/EFHOBZ). In comparison, control cells transfected with nontargeting siRNAs had a mean SG score of 1.9.

To our surprise, phosphotransferases associated with cell cycle regulation, proliferation, or DNA damage were highly represented among the candidates (35%; Fig. 1 C). Those associated with immunity and inflammation (12%) or carbohydrate metabolism (10%) were also abundant, whereas only a few candidates were associated with ribosome and ribonucleotide biogenesis (3%). Of 15 top candidates chosen for validation, KD of N-acetylglucosamine kinase, ROS proto-oncogene 1, polynucleotide kinase-phosphatase, and CDK1 reproducibly resulted in SG assembly in >5% of the cells (Fig. 2 A). Given its central role for mitotic entry and its general importance in the cell cycle (Itzhaki et al., 1997; Santamaría et al., 2007), we decided to pursue CDK1 as a candidate that may connect proliferation rates with global protein synthesis.

### Inhibition of CDK1 reduces protein synthesis

Since SG-based screens report not only on regulators of translation but also on downstream factors that control the assembly of SGs, it was important to test if CDK1 influences global translation rates. To this end, we performed polysome profile analysis in HeLa cells subjected to nontargeting control, CDK1, or CDK2 KD for 72 h. An approximately twofold decrease in the percentage of polysomal ribosomes, which reflects the proportion of ribosomes engaged in translation, was observed upon CDK1 KD but not CDK2 KD (Fig. 2 B). Likewise, CDK1 KD led to an ~40% reduction in polypeptide synthesis as quantified by puromycin incorporation in single cells (Fig. 2 C). The single-cell analysis also revealed a higher degree of variability within the population of CDK1 KD cells, which might reflect cell-to-cell differences in KD efficiency despite the good overall KD efficiency of CDK1 and CDK2 as measured by Western blot analysis (Fig. 2 D). In addition, long-term consequences of CDK1 KD on cell cycle distribution may contribute to the observed variability in puromycin incorporation.

To focus on the direct, short-term consequences of CDK1 inhibition (CDK1i), we treated HeLa cells with the selective, ATP-competitive CDK1 inhibitor Ro3306 (Vassilev et al., 2006). CDK1i by Ro3306 treatment also led to the assembly of SGs (Fig. 3 A), to a gradual reduction in the rate of polypeptide synthesis (Fig. 3 B), and to a progressive decrease in the percentage of polysomal ribosomes, with a pronounced effect observed as early as 1 h after treatment (Fig. 3 C). These results could be further confirmed using a less selective CDK inhibitor, Roscovitine (Cicenas et al., 2015; Fig. 3 D). In addition, CDK1i reduced global translation rates not only in transformed cells (such as HeLa) but also in nontransformed primary mouse embryonic fibroblasts (MEFs; Fig. 3 E).

We then sought genetic evidence for a role of CDK1 in controlling protein synthesis. Since CDK1 is an essential gene, we made use of HT2-19, a human HT1080-derived cell line that contains one inactivated CDK1 allele, whereas the other allele is under control of a lac repressor and hence transcribed only in the presence of IPTG (Itzhaki et al., 1997). CDK1 levels were reduced at least twofold in HT2-19 cells cultured in the presence of IPTG compared with the parental HT-1080 cells, and polysomal ribosomes decreased from 43% to 32% (Fig. 3 F). CDK1 became barely detectable when HT2-19 cells were kept in the absence of IPTG for 7 d, and polysomal ribosomes decreased further to 18% (Fig. 3 F). These cells did not divide anymore but increased in cell size (arrows in Fig. S1 A).

### CDK1 controls global translation in a cell cycle–independent manner

CDK1 activity changes throughout the cell cycle: it starts to increase during S phase, reaches its maximum in metaphase, and declines rapidly in anaphase (Bashir and Pagano, 2005). In line with its activity profile, CDK1 was shown to control translation during mitosis at the level of translation initiation via phosphorylation of raptor (Ramírez-Valle et al., 2010), 4EBP1 (Heesom et al., 2001; Miettinen et al., 2019; Shuda et al., 2015), S6 kinase 1 (S6K1; Papst et al., 1998; Shah et al., 2003), and eIF4GI (Dobrikov et al., 2014), as well as at the level of elongation via phosphorylation of eEF1B (Monnier et al., 2001; Sivan et al., 2011) and eEF2K (Smith and Proud, 2008).

We noted that CDK1i led to SG formation in only ~10% of cells, which might be related to the peak of CDK1 activity in

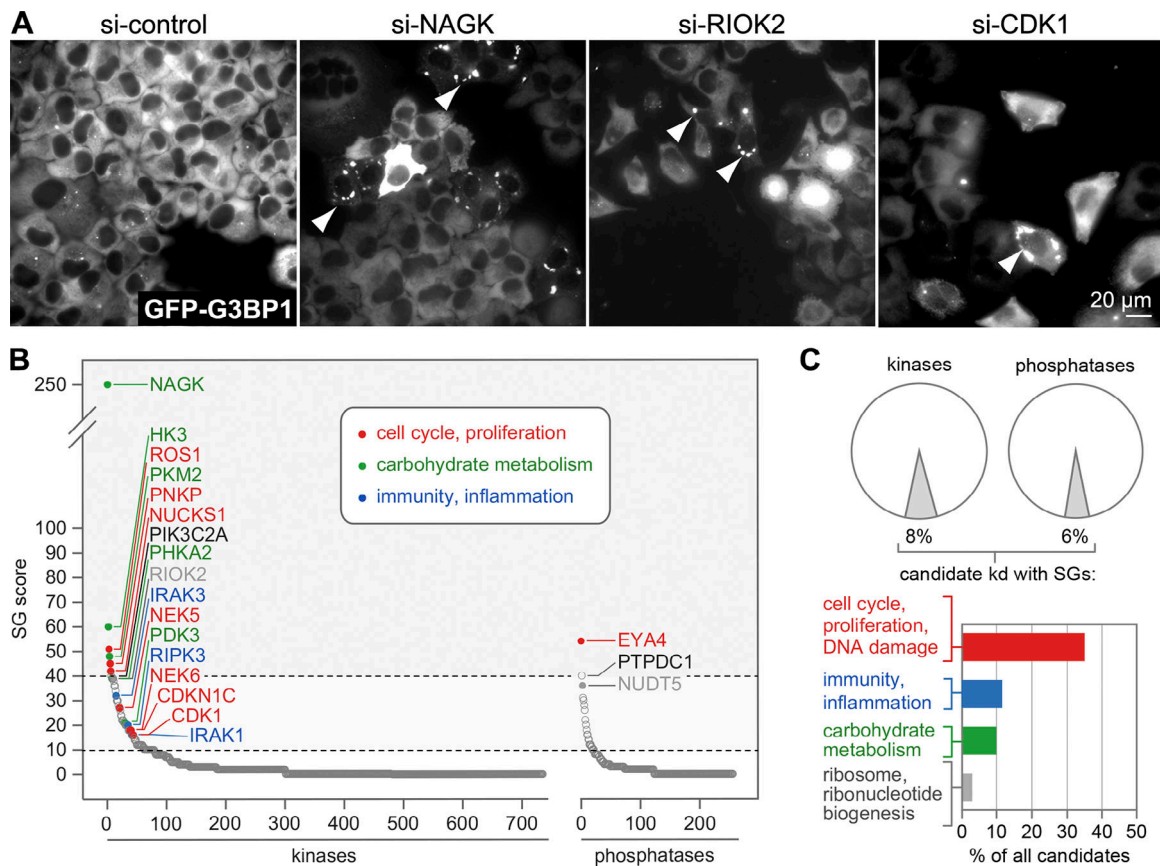

Figure 1. **SG assembly screen under regular growth conditions. (A)** The assembly of SGs was monitored in HeLa cells stably expressing GFP-G3BP1 following KD of 711 kinases and 256 phosphatases. Cells were transfected with four individual siRNAs per gene and 72 h later fixed for fluorescence microscopy. Representative images of the screen are shown; arrowheads indicate SG-containing cells; scale bar = 20 μm. **(B)** The screen was analyzed by calculating a SG score for each kinase/phosphatase KD, and the result was depicted by sorting all KDs according to their SG scores. Candidate kinases/phosphatases were identified by a SG score >10 (with two or more siRNAs) or >40 (with one siRNA); some of the candidates are labeled in the graph. **(C)** The graph depicts cellular functions highly represented among the candidate kinases/phosphatases, based on functional annotation in NCBI Gene and Uniprot databases.

mitosis. SGs appeared only upon long-term (16- or 24-h) inhibition of CDK1 concomitant with a strong cell cycle arrest at the G2/M boundary, whereas short-term (1–4-h) treatment with Ro3306 did not induce SG formation or alter the overall cell cycle profile (Figs. 4 A and S1 B). To test if SG formation upon CDK1i is restricted to a specific phase of the cell cycle, we made use of the FUCCI system and applied Ro3306 to HeLa cells stably expressing either Kusabira-Orange-Cdt1 or mVenus-Geminin (Sakaue-Sawano et al., 2008). While Cdt1 is expressed during G1- and early S phase, Geminin is expressed in S phase, G2 phase, and mitosis. The accumulation of Geminin-positive FUCCI cells confirmed the arrest in G2 phase upon treatment with Ro3306 for 16 h (Fig. S1 C). However, quantification of SG-positive cells revealed no preference for a particular cell cycle phase, since 10% of the Kusabira-Orange-Cdt1-positive cells and 9% of the mVenus-Geminin-positive cells contained SGs upon CDK1i (Fig. 4, B and C). In contrast, DMSO-treated FUCCI cells did not assemble any SGs (Fig. S1 C). This result suggested that CDK1 enhances global protein synthesis in a cell cycle phase–independent manner.

To further explore this possibility, we tested nonproliferating RPE1 cells after 48 h of serum starvation. The cells had entered

G0 phase, visible through the appearance of primary cilia (Fig. S1 D), and still responded to CDK1i by a strong reduction of their translation rate (Fig. 4 D). We also arrested HeLa cells at the G1/S boundary using a double thymidine (TT) block and, without release from the block, subjected them to CDK1i. Compared with asynchronously proliferating cells (with 70% polysomal ribosomes; Fig. 3 C), the cell cycle arrest alone led to a reduction of global protein synthesis (41% polysomal ribosomes), and treatment with Ro3306 for 4 h caused a further decrease to 30% polysomal ribosomes (Fig. 4 E). When we released the cells from the TT block into the different phases of the cell cycle, treatment with Ro3306 led to a similar reduction in the percentage of polysomal ribosomes 2 (S phase), 6 (late S/G2/M phase), 13 (late M/G1 phase), or 15 h (mostly G1 phase) after release (Fig. 3 F). Cell cycle phase and synchronicity were monitored by measuring Cyclin A, B, and D levels and histone H3 S10 phosphorylation (Fig. 4 G), as well as by recording cell cycle profiles using propidium iodide (PI) staining (Fig. 4 H). Importantly, translation was perturbed only minimally by the synchronization procedure, as cells after release from the block and untreated HeLa cells showed similar translation rates (compare 0 h Ro3306 in Figs. 3 C and 4 F). Moreover, CDK1i was applied for a short

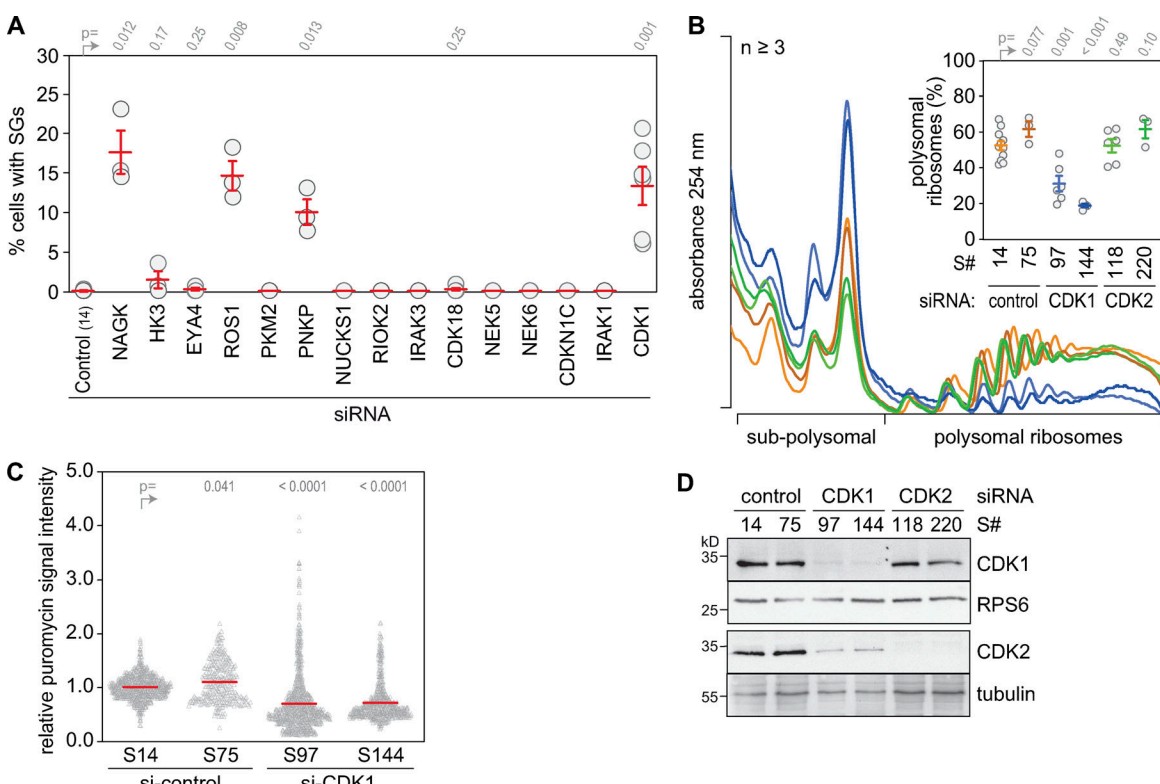

**Figure 2. CDK1 KD reduces global translation. (A)** The assembly of SGs was monitored in HeLa cells following KD of 15 candidate kinases/phosphatases. Cells were transfected with nontargeting control siRNA or one siRNA per gene. 72 h later, cells were fixed, and SG assembly was analyzed by IF microscopy of G3BP1. The percentage of cells with SGs was calculated (mean ± SEM, $n \geq 3$). **(B)** Polysome profiles from nontargeting control, CDK1, and CDK2 KD HeLa cells were recorded after sucrose density gradient centrifugation; the percentage of polysomal ribosomes is represented in the inset (mean ± SEM, $n \geq 3$). Statistical significance in A and B was determined by one-tailed Welch's $t$ test. **(C)** Puromycin incorporation signal intensities were detected by IF microscopy of fixed cells stained with anti-puromycin antibody. Values were calculated relative to nontargeting siRNA controls (mean ± SEM, $n \geq 2$). Statistical significance was determined by one-tailed ratio paired $t$ test. **(D)** CDK1 and CDK2 expression was assessed by Western blot analysis; RPS6 and tubulin serve as loading controls.

period of 2 h, which has only a marginal effect on the overall cell cycle profile (Fig. S1 B). Taken together, these experiments led us to conclude that enhancing protein synthesis is an extramitotic function of CDK1, which likely serves as a means to adjust protein synthesis to the overall proliferation rate rather than to a specific phase of the cell cycle.

### eIF2α phosphorylation, 4EBP1/2, and S6K1 signaling moderately contribute to translation control by CDK1

We then sought to explore the signaling pathway by which CDK1 controls protein synthesis. Various types of stress cause suppression of translation initiation via phosphorylation of eIF2α at serine (S)51, which prevents recharging of the initiator eIF2-GTP-tRNA$_i^{Met}$ ternary complex (Jackson et al., 2010). Western blot analysis showed robust phosphorylation of eIF2α 16–28 h after Ro3306 treatment (Fig. 5, A and B), whereas the onset of translation suppression was visible already 1 h after CDK1i (Fig. 3, B and C). Translation suppression upon Ro3306 treatment was partially impaired in MEFs containing a biallelic phosphodeficient eIF2α-S51A (AA) mutation (Scheuner et al., 2001) compared with MEFs expressing WT eIF2α-S51 (SS) alleles (Figs. 5 C and S2, A and B), indicating that eIF2α phosphorylation is alone not responsible for, but may contribute weakly to, translation inhibition after CDK1i.

Next, we examined targets of the mTOR pathway. 4EBP1, a direct target of mTORC1, showed a change in the phosphorylation pattern upon CDK1i and accumulated in a hypophosphorylated form 16 h after Ro3306 treatment (Fig. 5, D and E). Since 4EBP1 phosphorylation controls the integrity of the cap-binding complex (Sonenberg and Hinnebusch, 2009), we performed cap pulldown experiments using 7-methyl-GTP agarose beads. As expected, inhibition of mTORC1 using Torin1 (Thoreen et al., 2009) led to dissociation of eIF4G, eIF4A1, and eIF3B from eIF4E (Fig. 5, F–H). Treatment with Ro3306 for 4 or 16 h, however, did not interfere with the integrity of the cap-binding complex (Fig. 5, F–H), indicating that CDK1i does not repress translation via inhibition of eIF4G binding to eIF4E. Accordingly, we found that CDK1i reduced translation also in 4EBP1/2 double knockout (KO) cells (Le Bacquer et al., 2007), albeit to a slightly smaller extent than in 4EBP1/2 WT cells (Fig. 6, A and B, and Fig. S2 C).

To further explore the involvement of mTOR signaling in CDK1-dependent translational control, we analyzed MEFs overexpressing either WT mTOR or hyperactive mTOR mutants derived from renal cancer (Xu et al., 2016). The C1483F mutant caused pronounced phosphorylation of S6K1 under regular growth conditions (Fig. 6 C) and prevented dephosphorylation of S6K1 as well as hypophosphorylation of 4EBP1 upon CDK1i

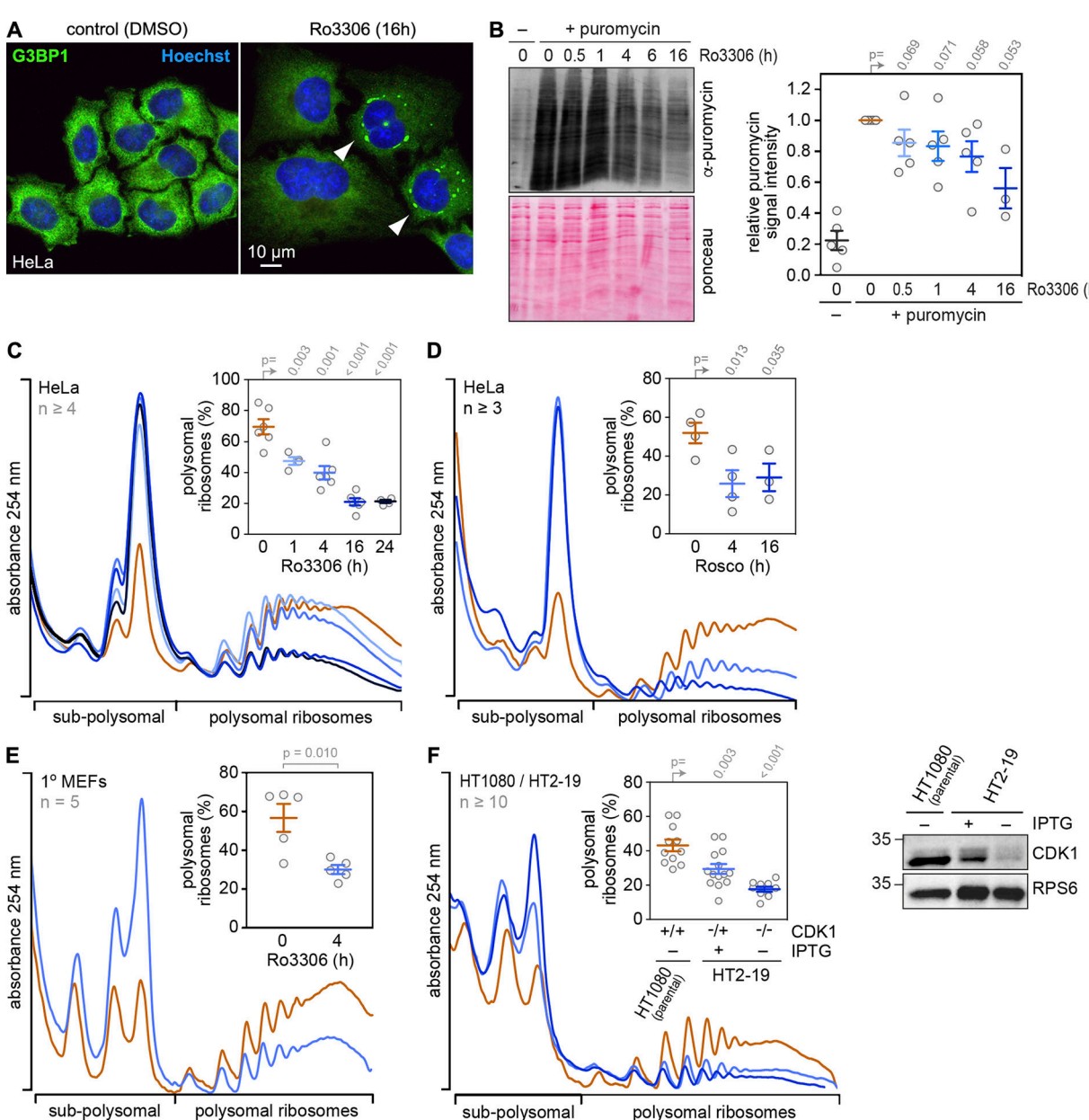

Figure 3. **Global translation suppression after pharmacological or genetic CDK1i. (A)** HeLa cells were treated with either solvent (DMSO) or the CDK1 inhibitor Ro3306 (10 µM) for 16 h. SG formation was analyzed by IF microscopy of fixed cells stained with anti-G3BP1 antibody and Hoechst; scale bar = 10 µm. **(B)** Incorporation of puromycin into nascent polypeptides was analyzed by SDS-PAGE and Western blotting from DMSO- or Ro3306-treated HeLa cells. Puromycin-labeled polypeptides were detected with anti-puromycin antibody; Ponceau staining served as loading control. One DMSO-treated sample was not incubated with puromycin and served as a negative control. Puromycin incorporation signal intensities were normalized to the Ponceau staining, and values were calculated relative to DMSO-treated control samples (mean ± SEM, $n \geq 3$). Statistical significance was determined by one-tailed ratio paired $t$ test. **(C and D)** Polysome profiles from DMSO- or Ro3306-treated HeLa cells (C) and Roscovitine-treated HeLa cells (D) were recorded after sucrose density gradient centrifugation; the percentage of polysomal ribosomes is represented in the inset (mean ± SEM, $n \geq 4$). Statistical significance was determined by one-tailed Welch's $t$ test. **(E)** Polysome profiles from DMSO- or Ro3306-treated primary MEFs were analyzed as in C. Statistical significance was determined by paired, one-tailed Student's $t$ test. **(F)** HT1080 and HT2-19 cells were seeded at subconfluence and kept in the presence or absence of IPTG (0.2 mM) for 7 d. Polysome profiles were recorded and analyzed as in C (mean ± SEM, $n \geq 10$). Statistical significance was determined by one-tailed Welch's $t$ test. CDK1 expression in HT1080 and HT2-19 cells was assessed by Western blot analysis; RPS6 serves as loading control.

(Fig. 6 D). Interestingly, CDK1i reduced translation to a similar degree in parental, mTOR WT, and mTOR C1483F–expressing MEFs (Figs. 6 E and S2 D), suggesting that CDK1i does not repress translation via inhibition of mTOR signaling. In conclusion, it is possible that 4EBP1 and 2 contribute to a small degree to the observed effect of CDK1i on global translation, yet our results

clearly show that mTOR is not the major pathway downstream of CDK1 regulating protein synthesis.

RPS6, a direct target of S6K1 and indirect target of mTORC1, was found to be strongly dephosphorylated early upon CDK1i (Fig. 5, D and E). We therefore examined whether S6K1 mediates CDK1-dependent control of translation by generating HeLa cells

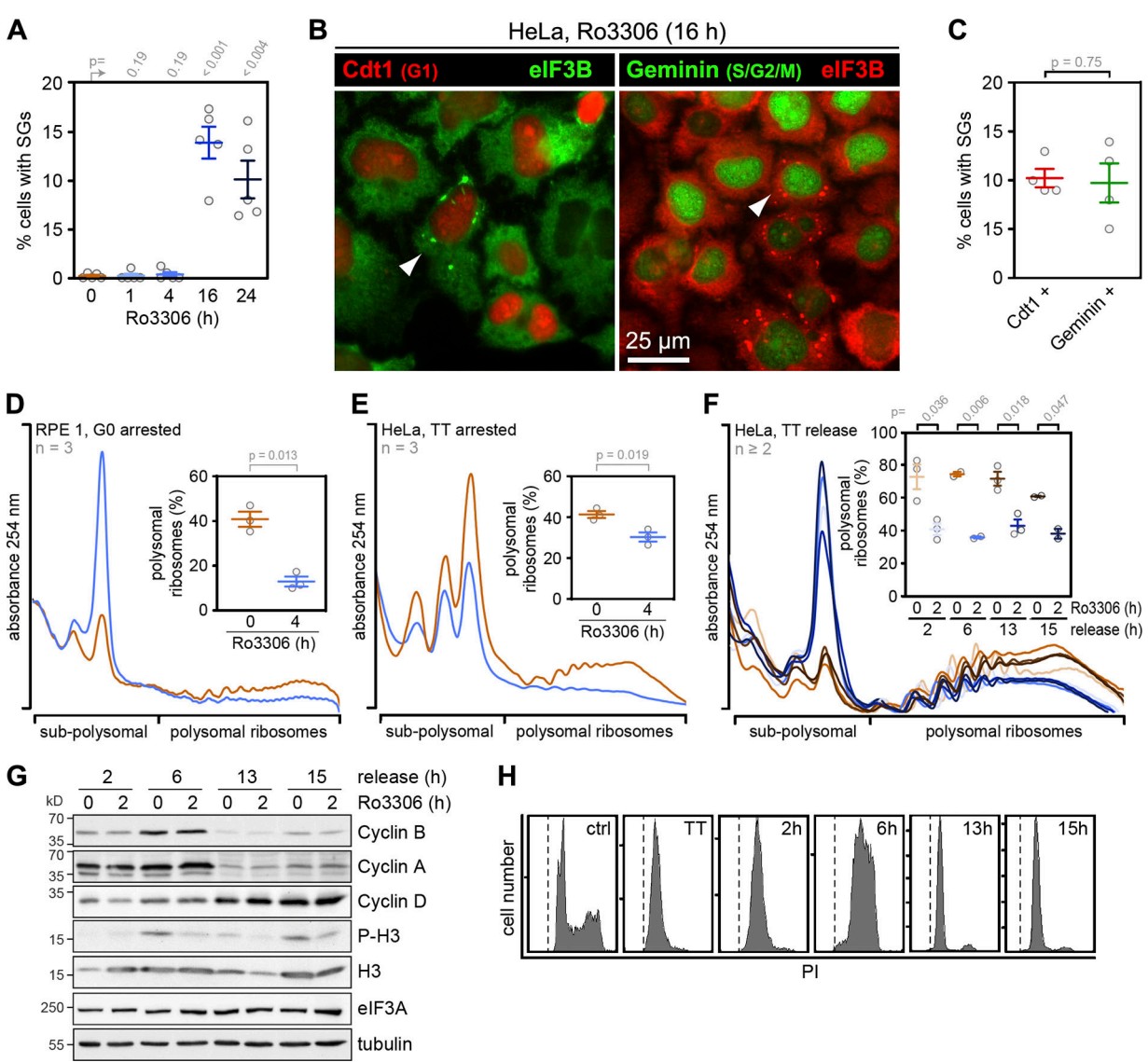

Figure 4. **Cell cycle phase–independent translation suppression upon CDK1i. (A)** HeLa cells were treated with either solvent (DMSO) or the CDK1 inhibitor Ro3306 (10 μM) for 1–24 h. SG formation was analyzed by IF microscopy of fixed cells stained with anti-G3BP1 and quantified (mean ± SEM, n = 5). Statistical significance was determined by paired, one-tailed Student's t test; scale bar = 25 μm. **(B)** HeLa FUCCI cells were treated with Ro3306 (10 μM) for 16 h, fixed, and analyzed for SG formation by IF microscopy upon staining with anti-eIF3B antibody. HeLa cells stably expressing Kusabira-Orange-Cdt1 (marker for G1 and early S phase) were used in the left panel; HeLa cells stably expressing mVenus-Geminin (marker for S, G2, and M phase) were used in the right panel. **(C)** Quantification of the percentage of SG-containing cells in Cdt1-positive (left) or Geminin-positive cells (right, n = 4). Statistical significance was determined by paired, two-tailed Student's t test. **(D)** RPE-1 cells were serum-starved for 48 h and subsequently treated with DMSO or Ro3306 (10 μM) for 4 h. Polysome profiles were recorded; the percentage of polysomal ribosomes is represented in the inset (mean ± SEM, n = 3). **(E)** HeLa cells were arrested at the G1/S boundary by a TT block and, without release from the block, treated with either solvent (DMSO) or Ro3306 (10 μM) for 4 h. Polysome profiles were analyzed as in D (mean ± SEM, n = 3). **(F)** HeLa cells were synchronized as in E and released from TT block for 2, 6, 13, or 15 h. Subsequently, cells were treated with DMSO or Ro3306 (10 μM) for 2 h, and polysome profiles were analyzed as in D (mean ± SEM, n ≥ 2). In D and E, statistical significance was determined by one-tailed paired Student's t test, and in F by one-tailed Welch's t test. **(G)** Expression of cyclins and the phosphorylation status of histone H3 (S10) from cells analyzed in F were assessed by Western blot analysis; eIF3A and tubulin levels serve as loading controls. **(H)** Cell cycle profiles from cells in F were analyzed by FACS using PI staining.

that stably overexpress WT or constitutively active (CA) S6K1 (Schalm et al., 2005). Phosphorylation levels of RPS6 were partially restored in the S6K1-overexpressing cells treated for 4 h with Ro3306 (Fig. 6 F), and translation suppression upon CDK1i was slightly, although significantly, reduced in comparison to control HeLa cells (Figs. 6 G and S2 E). We then tested whether RPS6 phosphorylation is responsible for this effect. In

MEFs expressing biallelic phosphodeficient RPS6$^{P-/-}$ (Ruvinsky et al., 2005), Ro3306 treatment suppressed translation to the same degree as in control RPS6$^{P+/+}$ MEFs (Figs. 6 H and S2 F). Taken together, these results indicated that eIF2α phosphorylation, 4EBP1/2, and S6K1 activity moderately contribute to translation control by CDK1, whereas RPS6 phosphorylation and mTOR activity are not directly involved.

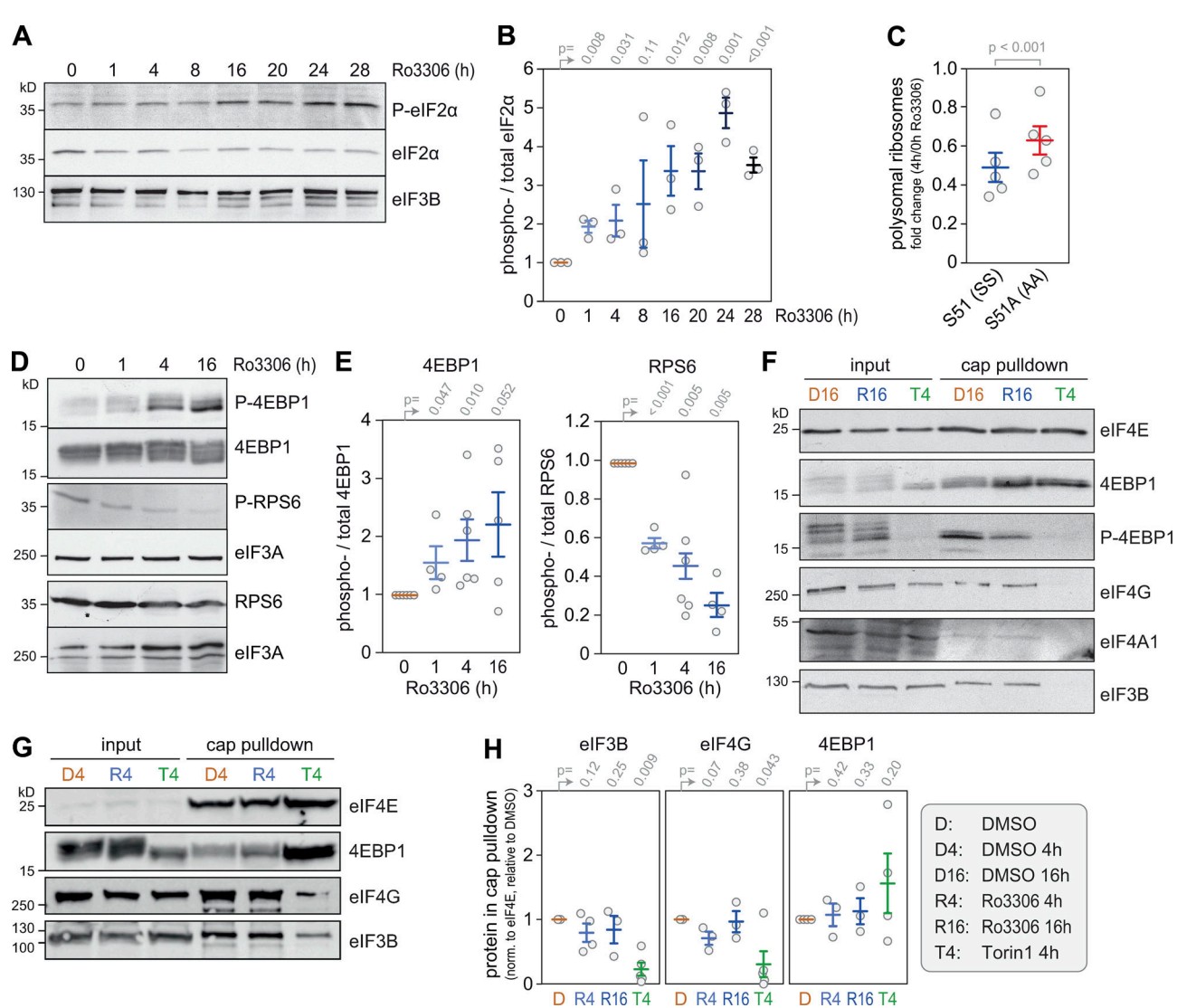

Figure 5. **Pathways signaling translational control downstream of CDK1i. (A)** Protein lysates were prepared from HeLa cells treated with solvent (DMSO, 24 h) or Ro3306 (10 μM, 1–28 h), and P-eIF2α (S51) was analyzed by Western blot analysis. **(B)** Quantification from Western blot analyses as shown in A (mean ± SEM, n = 3). Statistical significance was determined by one-tailed ratio paired t test. **(C)** The fold change in polysomal ribosomes (4 h Ro3306/DMSO control) was calculated based on polysome profiles recorded from SS and AA MEFs (mean ± SEM, n = 5). Statistical significance was determined by paired one-tailed Student's t test. **(D)** Protein lysates were prepared from HeLa cells treated with solvent (DMSO) or Ro3306 (10 μM). **(E)** The phosphorylation status of RPS6 (S235/S236) and 4EBP1 (T37/T46) was analyzed by Western blot analysis and quantified as in B. **(F and G)** Cytoplasmic lysates from DMSO-, Ro3306-, and Torin1-treated HeLa cells were subjected to cap pulldown experiments using 7-methyl-GTP agarose beads. Cap-associated factors eIF4E, 4EBP1, P-4EBP1 (T37/T46), eIF4G, eIF4A1, and eIF3B were detected by Western blot analysis. **(H)** For quantification of cap-associated factors, the amount of eIF3B, eIF4G, and 4EBP in the cap pulldown was normalized to the amount of precipitated eIF4E and is depicted relative to the value obtained for the DMSO control (mean ± SEM, n ≥ 3). Statistical significance was determined by one-tailed ratio paired t test.

## CDK1 affects phosphorylation of translation-associated factors

CDK1 was recently detected in a ribosome interaction capture mass spectrometry analysis (Simsek et al., 2017) and found to phosphorylate RPL12 (Imami et al., 2018). Together with our observation that CDK1i affects RPS6 phosphorylation (Fig. 5, D and E), these findings prompted us to explore whether CDK1 might influence more generally the phosphorylation of RPs and/or ribosome-associated factors. Polysome profile analysis revealed that a small proportion of CDK1 comigrates with polysomes and shifts to lighter fractions upon disassembly of polysomes by RNase treatment (Figs. 7 A and S3 A).

We then sought to identify possible targets of CDK1 associated with ribosomes using stable isotope labeling with amino acids in cell culture (SILAC)-based phosphoproteomics. Ribosomal fractions were obtained through sucrose gradient centrifugation from HeLa cells treated with either DMSO or Ro3306 for 4 h and subjected to mass spectrometry analysis. Phosphopeptide enrichment was achieved using either PhosSelect iron affinity gel IMAC beads (Repository Table R3, https://doi.org/10.11588/data/EFHOBZ) or TiO$_2$-based enrichment (Repository Table R4, https://doi.org/10.11588/data/EFHOBZ). Both analyses revealed a reduction in phosphorylation of RPS6, La ribonucleoprotein domain family member 1 (LARP1), death-associated

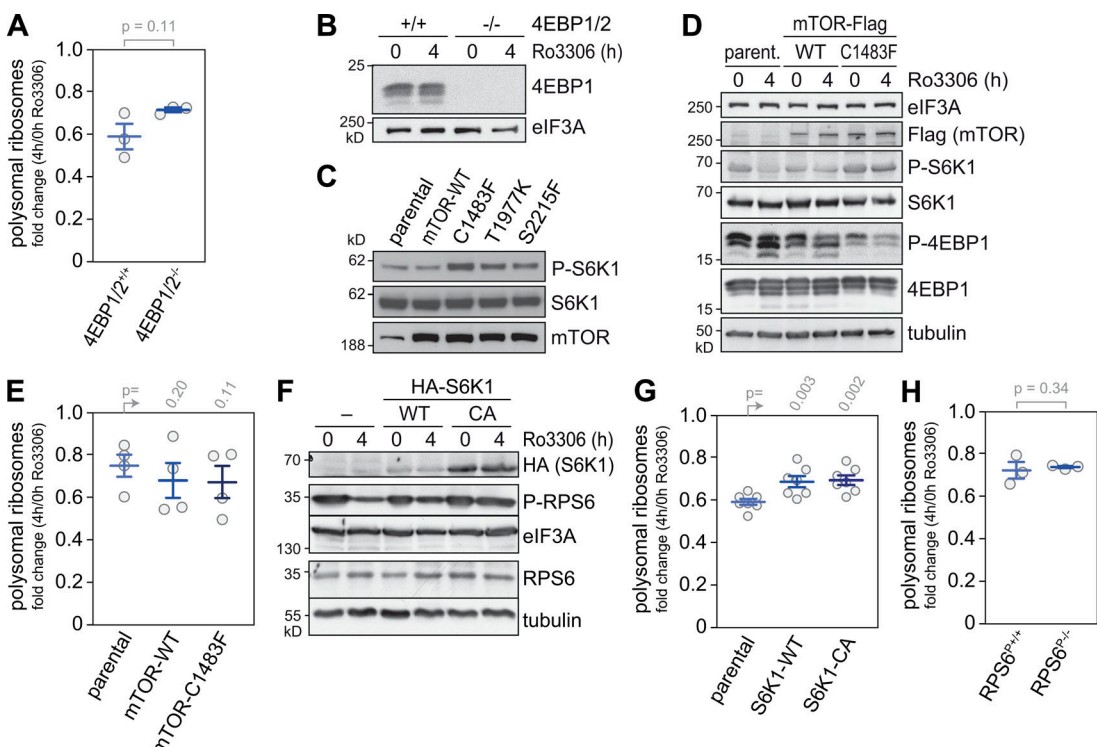

Figure 6. **Translational control downstream of CDK1i is independent of mTOR. (A)** The fold change in polysomal ribosomes (4 h Ro3306/DMSO control) was calculated based on polysome profiles recorded from 4EBP1/2[+/+] and 4EBP1/2[−/−] MEFs (mean ± SEM, n = 3). **(B)** 4EBP1 expression was assessed by Western blot analysis; eIF3A serves as loading control. **(C)** Doxycycline-dependent expression was induced for 20 h in SV40-immortalized K-Ras[G12D] MEFs (parental) and corresponding MEFs overexpressing mTOR-WT or the hyperactive mTOR mutants C1483F, T1977K, and S2215F. The phosphorylation status of p70 S6K1 (T389) and the expression levels of mTOR were analyzed by Western blot analysis. **(D)** Expression of mTOR-WT or hyperactive mTOR-C1483F was induced by treatment of SV40-immortalized K-Ras[G12D] MEFs with doxycycline for 20 h, followed by treatment with DMSO or Ro3306 (10 µM) for 4 h. The phosphorylation status of p70 S6K1 (T389) and 4EBP1 (T37/T46) was analyzed by Western blot analysis. Tubulin and eIF3A serve as loading controls. **(E)** The fold change in polysomal ribosomes from cells treated as in D was calculated (mean ± SEM, n = 4). **(F)** HeLa cells (control) and HeLa cells overexpressing HA-S6K1-WT or HA-S6K1-CA were treated with DMSO or Ro3306 (10 µM) for 4 h. The levels of S6K1 overexpression and the phosphorylation status of RPS6 were assessed by Western blot analysis; eIF3A and tubulin levels serve as loading controls. **(G)** The fold change in polysomal ribosomes was determined from cells in F (mean ± SEM, n = 7). **(H)** The fold change in polysomal ribosomes was determined from RPS6 WT (RPS6[P+/+]) and RPS6 phosphodeficient S235A, S236A, S240A, S244A, and S247A (RPS6[P−/−]) MEFs (mean ± SEM, n = 3). Statistical significance in A, E, G, and H was determined by paired one-tailed Student's t test.

protein, and Lamin A in response to CDK1i (Fig. 7 B). With the PhosSelect iron affinity gel IMAC-based method, which was more sensitive, additional Ro3306-sensitive sites were detected in several RPs (RPS6, RPS10, RPS17, RPL12, and RPL29), translation factors (eIF2B4, eIF3 subunits, eIF4A1, eIF4B, eIF4GI, and eIF5B), and translation regulators (LARP1 and YTH N[6]-methyladenosine RNA binding protein 1; Fig. S3 B, Repository Table R3).

**CDK1 strongly enhances 5′TOP mRNA translation via LARP1**

To gain further insight into the translation regulatory function of CDK1, we next performed ribosome footprint analysis (Ribo-Seq). Cell cycle phase–dependent effects were avoided by using RPE1 cells arrested in G0 through serum starvation, and ribosome density (RD) was measured at an early time point (4 h) after CDK1i. As an internal standard, equal amounts of a yeast lysate were spiked into the RPE1 cell lysates before RNase I digestion, which allowed us to assess both the global and transcript-specific effects of Ro3306 on translation. To reduce distortion of results through ligation biases, the input RNA was fragmented by alkaline hydrolysis and subjected to the same library preparation protocol

as the ribosomal footprints. Quality assessment showed the desired read lengths (Fig. S4 A), pronounced periodicity, and ORF enrichment for the footprints, but not the input RNA (Fig. S4 B), as well as adequate reproducibility between biological replicates (Fig. S4 C). As expected, CDK1i led to a global drop in RD (Fig. 8 A, most transcripts below the diagonal), resulting in a twofold reduction of the mean RD (Fig. 8 B). This result is in good agreement with the threefold reduction in polysomal ribosomes measured by polysome profiling (Fig. 4 D).

The analysis of individual transcripts revealed that CDK1i causes pronounced suppression of 5′TOP mRNAs, which includes all mRNAs encoding cytosolic RPs (Fig. 8, A and C; a table with all values is available in the GEO database, accession no. GSE128538). In contrast, mRNAs encoding mitochondrial RPs, which do not contain a 5′TOP motif, or internal ribosome entry site (IRES)-containing mRNAs, were not particularly sensitive to CDK1i (Fig. 8, A and C). We confirmed in HeLa cells that Ro3306 treatment preferentially reduces polysome association of 5′TOP mRNAs (RPLP0 and PABPC4), whereas control mRNAs (EIF2S1 and NCL) were barely affected (Fig. 8 D, repeats shown in Fig. S5, A and B).

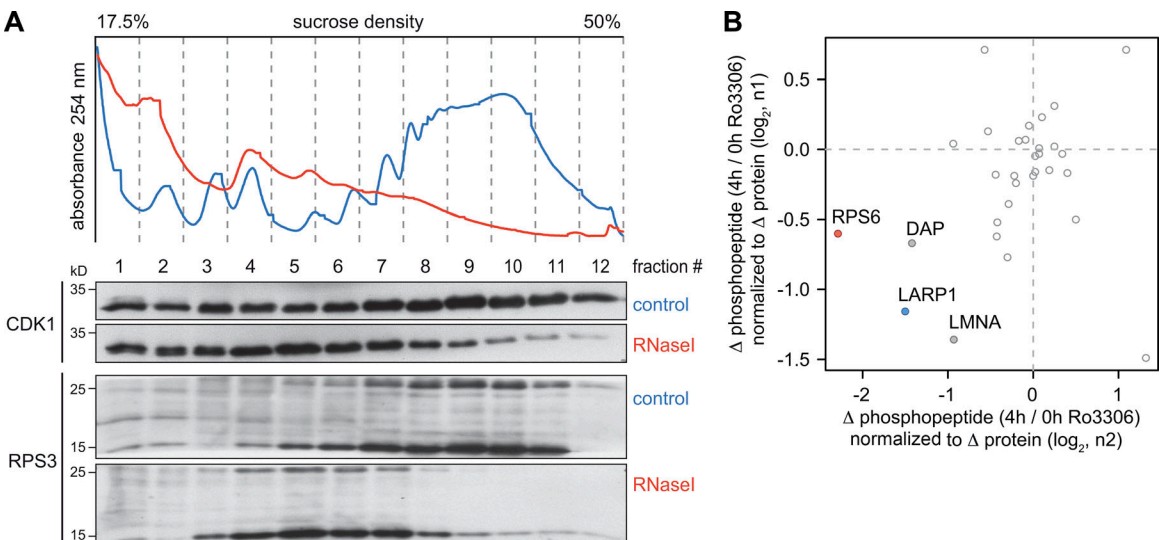

**Figure 7. CDK1-dependent phosphorylation events associated with ribosomes. (A)** HeLa cell lysates, either untreated or subjected to RNase I digestion, were fractionated following sucrose density gradient centrifugation. Association of CDK1 and RPS3 with the different fractions was monitored by Western blot analysis. **(B)** For phosphoproteomics of ribosomal fractions, HeLa cells were SILAC labeled and treated with either DMSO or Ro3306 for 4 h. After lysis and disassembly of polysomes in low-magnesium buffer, samples were mixed, and ribosomal fractions obtained by sucrose density centrifugation were subjected to phosphopeptide enrichment followed by mass spectrometry and MaxQuant analysis. For all phosphopeptides detected under both conditions, the ratio (Δ phosphopeptide abundance, 4 h/0 h Ro3306) was normalized to the ratio of the corresponding total protein (Δ protein abundance, 4 h/0 h Ro3306). Normalized ratios from the first repeat (n1) were then plotted against the second repeat (n2). Phosphopeptides derived from LARP1 (blue) and RPS6 (red) are color coded.

5′TOP mRNA translation was shown to be controlled by LARP1, which directly competes with eIF4E for binding to the cap of these transcripts (Lahr et al., 2017). Although LARP1 was initially reported to enhance translation of 5′TOP mRNAs under normal growth conditions (Tcherkezian et al., 2014), more recent evidence suggests that LARP1 represses 5′TOP mRNA translation (Fonseca et al., 2015; Philippe et al., 2018). This switch in activity appears to be controlled by phosphorylation of LARP1 (Hong et al., 2017). Since our phosphoproteomics analysis had indicated prominent changes in LARP1 phosphorylation (Figs. 7 B and S3 B), we decided to test whether the inhibitory effect of CDK1i on 5′TOP mRNA translation was dependent on LARP1 using KO cells (Fonseca et al., 2018; *Preprint*). Whereas CDK1i led to a strong reduction of 5′TOP mRNA association with polysomes in WT HEK293T cells, the effect was abolished in HEK293T LARP1⁻/⁻ cells (Fig. 8, E and F, repeats shown in Fig. S5, C and D). Notably, reexpression of LARP1 in LARP1⁻/⁻ cells (Fig. 8 E) restored the inhibitory effect of CDK1i on polysome association of 5′TOP mRNAs (Figs. 8 F and S5, C and D). Since both HEK293T WT and LARP1⁻/⁻ cells responded to CDK1i by a reduction of polysomes (Fig. S5 E), we concluded that LARP1 is not linked to the effect on global protein synthesis, while it is necessary for the repression of 5′TOP mRNA translation upon CDK1i. mTOR-dependent phosphorylation was shown to reduce binding of LARP1 to 5′TOP mRNAs (Philippe et al., 2018). In contrast, inhibition of CDK1 did not affect binding of 5′TOP mRNAs (RPL29 and RPL32) to LARP1, as determined by RNA immunoprecipitation (RNA-IP; Fig. S5 F), suggesting that CDK1 regulates LARP1 activity through other means than RNA binding and independently of mTOR.

## Discussion

Early experiments measuring the incorporation of radiolabeled nucleosides and amino acids had already pointed to a tight connection between the proliferation rate and the rate of protein synthesis in cultured fibroblasts subjected to contact inhibition (Levine et al., 1965) or serum deprivation (Rudland, 1974). Current concepts on mechanisms that couple the two rates focus on the mTOR signaling network, which integrates cues from growth factors and nutritional sensing to control a cell growth checkpoint in late G1 (Foster et al., 2010). The connection is based on the notion that active mTOR, among its many effector functions, promotes cell proliferation as well as protein synthesis and ribosome biogenesis (Laplante and Sabatini, 2012).

Our SG-based screen for potential activators of translation revealed several candidate kinases that are primarily associated with cell cycle, proliferation, and DNA damage (Fig. 1). A similar observation was made in an earlier screen by the Pelkmans laboratory, where inhibitors of several cell cycle kinases were found to prevent the dissolution of SGs (Wippich et al., 2013). These findings prompted us to test whether cell cycle kinases may be directly involved in controlling protein synthesis, and we decided to focus on CDK1 given its central role in driving the cell cycle (Santamaría et al., 2007).

Our analysis uncovered a cell cycle–independent function of CDK1 in enhancing overall protein synthesis. We found that global translation rates are strongly reduced upon KD (Fig. 2, B–D), pharmacological inhibition (Fig. 3, B–E), or genetic inactivation of CDK1 (Fig. 3 F). The effect was general as Ro3306 suppressed translation in both transformed (HeLa, HT1080, HEK293T, MEFs)

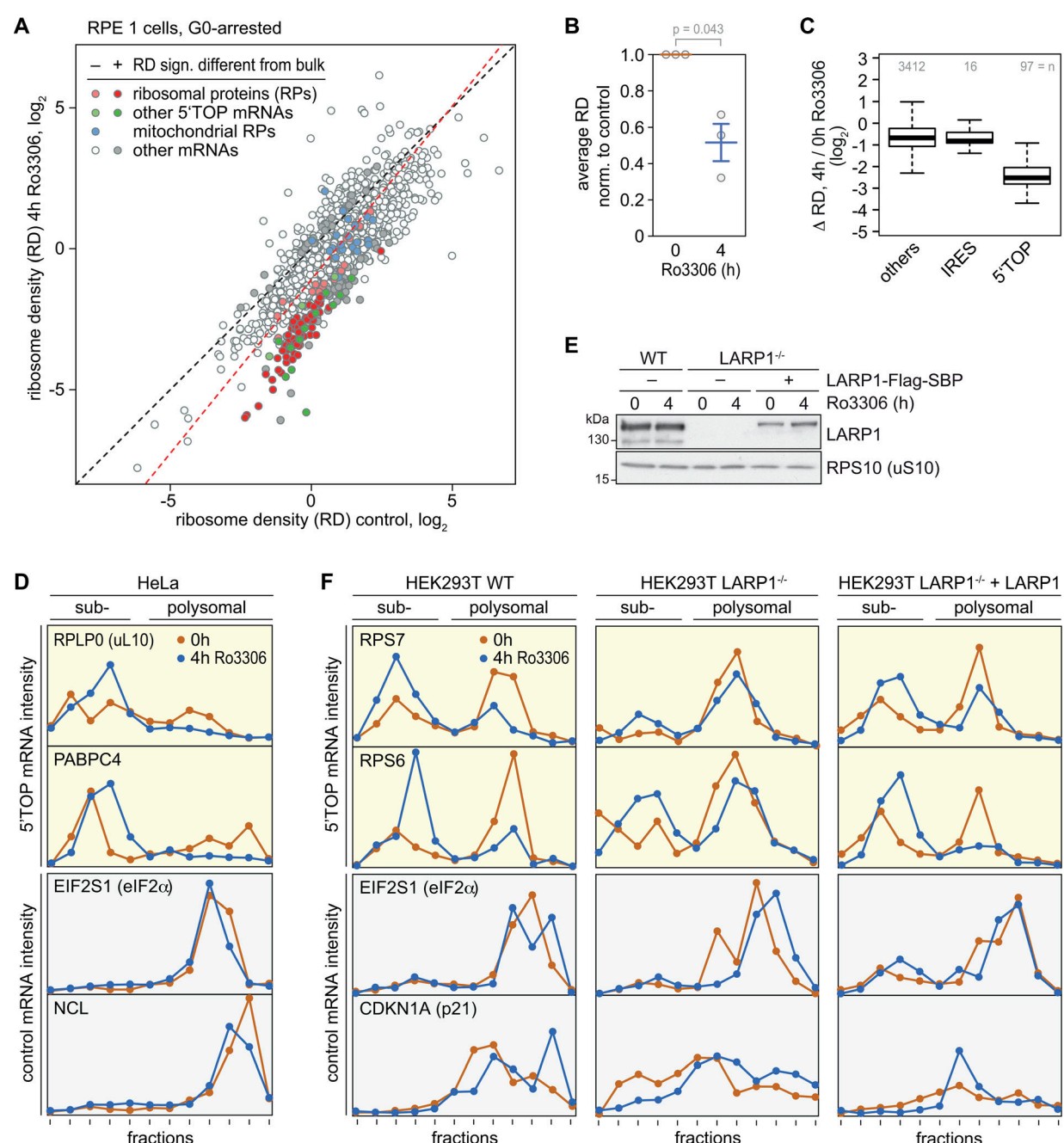

Figure 8. **LARP1-dependent suppression of 5′TOP mRNA translation upon CDK1i. (A)** For Ribo-Seq analysis, RPE1 cells were serum-starved for 48 h followed by a 4-h treatment with DMSO or Ro3306 (10 μM). Equal amounts of a yeast lysate were spiked into the DMSO- and Ro3306-treated samples. RDs (no. ribosome footprints/no. ORF-spanning reads in input RNA) were calculated after normalization to the yeast spike-in footprints from $n = 3$ biological repeat experiments. **(B)** Based on the Ribo-Seq analysis in A, the average RD was calculated after normalization (norm.) to the yeast spike-in. Statistical significance was determined by one-tailed ratio paired $t$ test. **(C)** Based on the Ribo-Seq analysis in A, the fold change in RD (Δ RD) was calculated for IRES-containing mRNAs, 5′TOP mRNAs, and all other mRNAs. **(D)** Polysome association of 5′TOP (RPLP0 and PABPC4) and ORF size-matched non-TOP (EIF2S1 and NCL) mRNAs was analyzed by polysome fractionation and subsequent qPCR analysis from DMSO- or Ro3306-treated (10 μM, 4 h) HeLa cells. **(E)** Western blot analysis of LARP1 expression in HEK293T WT cells, HEK293T LARP1$^{-/-}$ cells, and HEK293T LARP1$^{-/-}$ + LARP1 cells expressing LARP1-Flag-SBP. RPS10 serves as loading control. **(F)** Polysome association of 5′TOP (RPS6 and RPS7) and ORF size-matched non-TOP (EIF2S1, CDKN1A) mRNAs was analyzed by polysome fractionation and subsequent qPCR analysis from DMSO- or Ro3306-treated (10 μM, 4 h) HEK293T WT, LARP1$^{-/-}$, or LARP1$^{-/-}$ + LARP1 cells.

and nontransformed (RPE1, primary MEFs) cells (Fig. 3, B–F; Fig. 4, D–F; Fig. S2; and Fig. S5 E).

Previous studies identified several translation-associated factors as direct substrates of CDK1 in mitosis, including S6K1 (Papst et al., 1998; Shah et al., 2003), Raptor (Ramírez-Valle et al., 2010), eEF2K (Smith and Proud, 2008), eIF4GI (Dobrikov et al., 2014), 4EBP1 (Heesom et al., 2001; Shuda et al., 2015; Velásquez et al., 2016), eEF1D (Monnier et al., 2001; Sivan et al., 2011), and RPL12 (Imami et al., 2018). These studies suggested that CDK1 regulates translation during mitosis, though consequences

for the following interphase were also observed (Miettinen et al., 2019). In fact, many of the reported CDK1-dependent phosphorylation events (S6K1, eIF4GI, and eEF1D) were proposed to repress global translation to increase the translation of mitosis-specific transcripts. On the other hand, CDK1-dependent phosphorylation of 4EBP1 and eEF2K were linked to a positive role of CDK1 in global translation during mitosis. Our analyses suggest that CDK1 also exerts a translation regulatory function independently of the cell cycle phase, since Ro3306 treatment suppressed translation in cells arrested in G0 (Fig. 4 D) or at the G1/S boundary (Fig. 4 E), as well as in synchronized S-, G2/M-, or G1-phase cells (Fig. 4, F–H). Thus, we propose that in addition to controlling translation during mitosis, CDK1 serves as a relay to balance the overall proliferation rate of a cell with the overall protein synthesis rate. This may be linked to the notion that CDK1 phosphorylates different substrates depending on its activity during the cell cycle, its subcellular localization, its association with coactivators (cyclins or RINGO proteins), and possibly its phosphorylation status (Gupta et al., 2007; Hochegger et al., 2008; Nebreda, 2006; Swaffer et al., 2016).

When addressing the mechanism by which CDK1 enhances global protein synthesis, we found that CDK1 influences translation initiation via multiple, possibly redundant pathways. First, we found an increase in eIF2α phosphorylation upon CDK1i (Fig. 5, A and B), and since translation suppression was mildly reduced in MEFs expressing nonphosphorylatable eIF2α S51A (Fig. 5 C), one role of CDK1 could be to promote recharging of the eIF2-GTP-tRNA$_i^{Met}$ ternary complex. Second, we observed a pronounced reduction in RPS6 phosphorylation upon CDK1i (Fig. 5, D and E), and reduced translation suppression in HeLa cells overexpressing S6K1 (Fig. 6 G) indicated that S6K1 signaling also contributes to CDK1-dependent control of global translation. Third, we detected a change in the phosphorylation pattern of 4EBP1 upon CDK1i (Fig. 5, D and E), which is in agreement with earlier studies on CDK1-dependent phosphorylation of 4EBP1 (Heesom et al., 2001; Shuda et al., 2015; Sun et al., 2019). The importance of 4EBP1 phosphorylation on mitotic translation is discussed controversially. While some studies suggest that mitotic and/or CDK1-dependent phosphorylation of 4EBP1 replaces mTOR-dependent phosphorylation of 4EBP1 during mitosis (Miettinen et al., 2019; Shuda et al., 2015) and thus relieves the inhibitory effect of 4EBP1 on cap-dependent translation; others (Velásquez et al., 2016, Sun et al., 2019) do not observe a measurable difference in cap-dependent translation in the presence or absence of mitotic 4EBP1 phosphorylation, which is characterized by an additional phosphorylation event on S83. In our experiments, we measured only slightly reduced translation inhibition upon CDK1i in unsynchronized 4EBP1/2 KO cells (Fig. 6 A) and therefore suggest that 4EBP1 contributes to translation control downstream of CDK1 but cannot be the major translational target of CDK1 outside of mitosis. Of note, our results using the hyperactive mTOR mutant C1483F clearly indicate that CDK1 does not regulate translation via mTOR (Fig. 6 E). Also, CDK1i did not alter the integrity of the cap binding complex (Fig. 5, F–H), providing further evidence that CDK1 regulates translation independently of mTOR. Taking these results together,

it appears that eIF2α phosphorylation, S6K1, and 4EBP1/2 activity all contribute to translation control by CDK1, albeit only to a moderate degree.

Interestingly, we observed that a small proportion of CDK1 cosediments with polysomes (Fig. 7 A), and association of CDK1 with ribosomes was also reported by a mass spectrometry approach (Simsek et al., 2017). This is in line with the notion that RPL12 is a known substrate of CDK1, and RPL12 phosphorylation was recently shown to enhance a mitotic translation program (Imami et al., 2018). Hence, it is possible that CDK1 stimulates global translation by phosphorylating additional proteins of, or associated with, the ribosome.

Finally, our results show that CDK1 is a pronounced activator of 5′TOP mRNA translation, which includes the synthesis of all RPs (Fig. 8). Hence, CDK1 has a sustained effect on global protein synthesis in proliferating cells, as it enhances translation at the initiation level, and possibly also at the elongation level (Smith and Proud, 2008), and by promoting biogenesis of the protein synthesis machinery.

In agreement with the pronounced effect of CDK1 on 5′TOP mRNA translation, it is well known that cell cycle progression tightly correlates with 5′TOP mRNA translation. For example, cell cycle arrest in G0, at the beginning of S phase, or in M phase strongly reduces translation of 5′TOP mRNAs (Meyuhas and Kahan, 2015). Likewise, translation of 5′TOP mRNAs is low in resting adult liver cells but high in developing fetal liver cells, as well as in proliferating adult liver cells during regeneration (Aloni et al., 1992). Moreover, 5′TOP mRNAs were found to be resistant to mTOR inhibition in mitosis, but not in interphase (Sun et al., 2019), which would correlate well with the activity profile of CDK1. We propose that CDK1 has a central role in coupling 5′TOP mRNA translation with the proliferation status of the cell since (a) LARP1 phosphorylation is strongly dependent on CDK1 activity (Fig. 7 B), and (b) CDK1 controls 5′TOP mRNA translation in a LARP1-dependent manner (Fig. 8, D–F). Future studies will need to show if LARP1 is a direct target of CDK1 and address the detailed mechanism by which CDK1 antagonizes the inhibitory activity of LARP1 on 5′TOP mRNA translation.

Taken together, our results suggest that CDK1 acts as a central relay connecting proliferative cues with protein synthesis. This activity occurs in parallel with the mTOR kinase, which functions as a signaling hub that couples cues from growth factors and nutrient sensing with protein synthesis. CDK1 and mTOR thereby share common targets including S6K1, 4EBP1, and LARP1. Together with mTOR and Ras/Erk, CDK1 appears to form a homeostatic network that coordinates proliferative cues and growth signals with the availability of the protein synthesis machinery and the rate of protein synthesis.

## Materials and methods
### Plasmid generation
The GFP-G3BP1 sequence was obtained from J. Tazi (Institut de Génétique Moléculaire de Montpellier, Montpellier, France) and cloned into the NheI and EcoRI sites of pCI-puro, resulting in pCI-puro-GFP-G3BP1 (p2163). pKH3-HA-S6K1-WT (p2760) and pKH3-HA-S6K1-CA (F5A-T389E-R3A; p2762) were kindly

provided by J. Blenis (Weill Cornell Medicine, New York, NY). HA-S6K1-WT and HA-S6K1-CA sequences were amplified using oligonucleotides G4542 and G4543 from plasmid p2760 and p2762, respectively, and cloned into the SmaI sites of pWPI-BLR (Ruggieri et al., 2012), resulting in the generation of pWPI-BLR-HA-S6K1-WT (p3669) and pWPI-BLR-HA-S6K1-CA (F5A-T389E-R3A; p3671). The LARP1 (isoform 1) sequence was derived from cDNAs obtained from Sarah Blagden (University of Oxford, UK) and Bruno Fonseca (University of Ottawa, Canada) and cloned into the KpnI and NotI sites of pTOPuro-Flag-TEV-SBP (p3373), resulting in the generation of pTOPuro-LARP1-Flag-SBP (p3771). pWPI-FUCCI-Kusabira-Orange-Cdt1-Zeo and pWPI-FUCCI-mVenus-Geminin-Zeo were generated by EcoRI-XbaI excision of the human Cdt1 cDNA N-terminally fused to mKO2 from plasmids pCSII-EF-MCS-mKO2-hCdt1-(30/120) and of the human Geminin cDNA N-terminally fused to mVenus from pCSII-EF-MCS-mVenus-hGeminin-(1/110), respectively (both kindly provided by A. Miyawaki, RIKEN Center for Brain Science, Wako, Japan; Sakaue-Sawano et al., 2008). Both sequences were inserted into the lentiviral transduction vector pWPI carrying a zeocin resistance gene.

## Generation of stable cell lines and KO cell lines

Hela GFP-G3BP1 cells were generated by plasmid transfection of pCI-puro-GFP-G3BP1 (p2163) using polyethylenimine. 24 h after transfection, cells were subjected to selection by addition of 2 µg/ml puromycin (Gibco) and FACS sorted using a BD FAC-SAria IIIu cell sorter after 2 wk. HeLa-FUCCI-Kusabira-Orange-hCdt1 and HeLa-FUCCI-mVenus-hGeminin cells were generated by lentiviral transduction of pWPI-FUCCI-Kusabira-Orange-Cdt1-Zeo and pWPI-FUCCI-mVenus-Geminin-Zeo, respectively. Retroviral transduction and generation of stable cell lines was performed as described previously (Ruggieri et al., 2012). In short, HeLa cells were seeded into 6-cm-diameter dishes and transfected using the CalPhos mammalian transfection kit (Becton Dickinson) as recommended by the manufacturer. For transfection, the packaging plasmid (pCMVΔ8.91), the transfer vector (pWPI-based), and the vesicular stomatitis virus envelope glycoprotein expression vector (pMD2.G) were used in a concentration ratio of 3:3:1. Transduction of HeLa cells with the lentiviral particles was repeated three times every 12 h to achieve high integration numbers. Transduced cell pools were subjected to selection with 100 µg/ml zeocin (Invitrogen), and high-expressing cells were sorted by FACS. HeLa-HA-S6K1-WT and HeLa-HA-S6K1-CA cells were generated accordingly by retroviral transduction using pWPI-BLR-HA-S6K1-WT (p3669) and pWPI-BLR-HA-S6K1-CA (F5A-T389E-R3A; p3671). Cells were subjected to selection by addition of 5 µg/ml blasticidin (Invitrogen). Plasmids for inducible expression of mTOR mutants in a modified version of the retroviral vector pTRE-Tight were a gift from J. Hsieh (Cornell University, New York, NY; Xu et al., 2016). SV40-immortalized K-Ras^{G12D} MEFs (Tuveson, 2004) were first transduced with rtTA3 (pMSCV-rtTA3-PGK-hygro) through retroviral delivery, followed by selection with 100 µg/ml hygromycin. Selected cells were then transduced with mTOR mutants through retroviral delivery, followed by selection with puromycin at 2 µg/ml. Expression of mTOR mutants was induced by addition of 1 µg/ml doxycycline for 20 h. To create plasmids for expression of LARP1-specific gRNAs, LARP1-oligo1 and LARP1-oligo2 were annealed and cloned into Esp3I-digested LentiCRISPRv2, resulting in vector LentiCRISPR-LARP1gRNA. 1 d before transfection, HEK293T cells were seeded at a density of $3 \times 10^5$ cells/well in six-well plates. Transfections were performed using 1 µg LentiCRISPR-LARP1 gRNA and Lipofectamine 2000 Transfection Reagent (Invitrogen) according to the manufacturer's protocol. Single-cell clones were generated 2 d after transfection, and genomic DNA was purified using the GenElute Mammalian Genomic DNA Miniprep Kits (Millipore) according to the manufacturer's protocol. LARP1 KO was verified by PCR on genomic DNA using the primers LARP1-oligo3 and LARP1-oligo4, followed by Sanger sequencing of the resulting PCR product using the primer LARP1-oligo3. HEK293T LARP1^{-/-} cells were transfected with pTOPuro-LARP1-Flag-SBP (p3771) to generate LARP1^{-/-} + LARP1 cells. Cells were subjected to selection by addition of 2 µg/ml puromycin (Gibco), and single-cell clones were analyzed by Western blot and immunofluorescence (IF) microscopy for expression of LARP1-Flag-SBP.

## Cell culture

HeLa cells, HEK293T cells, and MEFs were maintained in DMEM (Gibco) containing 10% FCS (PAA Laboratories), 2 mM L-glutamine, 100 U/ml penicillin, and 100 µg/ml streptomycin (all PAN Biotech). SS (WT) and AA MEFs (Scheuner et al., 2001) were a kind gift from R. Kaufmann (Sanford Burnham Prebys Medical Discovery Institute, La Jolla, CA); RPS6^{P+/+} and RPS6^{P-/-} MEFs were generously provided by O. Meyuhas (Hebrew University of Jerusalem, Israel). HT1080 and HT2-19 cells (Itzhaki et al., 1997) were a kind gift from A. Porter (Imperial College School of Medicine, London, UK) and were maintained in DMEM high glucose and nonessential amino acids medium containing 10% FCS, 2 mM L-glutamine, 10 mM pyruvate, 40 U/ml penicillin, and 40 µg/ml streptomycin (all PAN Biotech). HT2-19 cells were additionally supplemented with 0.2 mM IPTG (AppliChem). For CDK1 depletion, HT2-19 cells were seeded at very low density and cultured in the absence of IPTG for 7 d. RPE1 cells, kindly provided by I. Hoffmann (German Cancer Research Center, Heidelberg, Germany), were cultured in Ham's F-12 medium (1:1; Millipore) containing 10% FCS, 2 mM L-glutamine, 100 U/ml penicillin, and 100 µg/ml streptomycin. HeLa-FUCCI-Kusabira-Orange-hCdt1, HeLa-FUCCI-mVenus-hGeminin, and HeLa-GFP-G3BP1 cells were FACS sorted, cultured without selection pressure, and maintained at low passage numbers. Primary MEFs prepared from 13- to 14-d-old embryos with C57BL/6N genetic background were a kind gift from F. Tuorto (German Cancer Research Center). The mouse husbandry and the experiment to isolate primary MEFs were performed at the German Cancer Research Center pathogen-free animal facility according to applicable laws and regulations. Primary MEFs were cultured in DMEM complete medium (Gibco) supplemented with 10% FBS (Gibco) and 1% penicillin-streptomycin antibiotic cocktail (Gibco). All cells were cultured at subconfluence, at 37°C in 5% $CO_2$. For treatment with inhibitors, cells were seeded the evening before, and Ro-3306 (Sigma-Aldrich or Merck Millipore, 10 µM), Roscovitine (Sigma-Aldrich, 20 µM), Torin-1 (200 nM,

Tocris Bioscience), or control solvent (DMSO) were diluted in fresh medium, which was added onto the cells for the indicated times. For synchronization, HeLa cells were subjected to a TT block following standard procedures (first block, 18 h, 2 mM thymidine; 9-h release; second block, 18 h, 2 mM thymidine).

## Screening approach and SG score

For the siRNA screen, 96-well MGB096-1-2-LGL matriplates (Brooks) were coated with an siRNA transfection mix containing the Dharmacon siGenome siRNA libraries GU-003505 Human Protein Kinase and GU-003705 Human Phosphatase from Thermo Fisher Scientific, Lipofectamine RNAiMAX (Invitrogen), 57 mM sucrose, 0.03% gelatin/fibronectin in solution, and OPTIMEM. The coated plates were prepared at the Cellnetworks Advanced Biological Screening Facility of Heidelberg University using the Hamilton STAR pipetting robot. The siRNA libraries were directed against 711 human kinases and 256 human phosphatases including four individual siRNAs per gene. By seeding 2,000 HeLa-GFP-G3BP1 cells per well into the coated 96-well plates, siRNAs were diluted to a final concentration of 50 nM, and KD was performed for 72 h. Cells were fixed for 10 min at RT using 4% PFA in PBS supplemented with Hoechst dye (1:10,000 diluted). Afterward, cells were washed three times and stored in PBS at 4°C in the dark until examination under the microscope. Seeding, washing, and fixation were done with a microplate suspensor (Thermo Fisher Scientific Multidrop Combi) to ensure fast, synchronous, and equal handling. SG formation was analyzed using a Nikon Eclipse Ti2-E microscope and a Nikon Plan Apo 60× oil objective (NA 1.4) that was constantly supplied with immersion oil by a pumping system. 16 images per well were taken automatically using a scientific complementary metal–oxide–semiconductor (sCMOS) camera (Flash4, Hamamatsu), Nikon JOBS software, and the Nikon perfect focus system, and images were subsequently analyzed by eye. For every phosphotransferase, an SG score was calculated by multiplying the sum of SG-containing cells, observed with all four siRNAs, by the number of siRNAs causing SGs.

## siRNA transfection

KD experiments were performed for 72 h using RNAiMAX Lipofectamine transfection reagent (Invitrogen) and reverse transfection according to the manufacturer's instructions. siRNAs were purchased from Eurofins MWG Operon; the S75 nontargeting control siRNA was purchased from Qiagen (10278210). All siRNAs were transfected at a final concentration of 50 nM. The following sequences were used: S14, 5′-GGUCCG GCUCCCCCAAAUG-3′ (C2, nontargeting); S97, 5′-GAUCAACUC UUCAGGAUUU-3′ (CDK1); S118, 5′-GAGCUUAACCAUCCUAAU A-3′ (CDK2); S144, 5′-GAUGUAGCUUUCUGACAAAAA-3′ (CDK1); S145, 5′-AAGAACCUACUUAAGAUAGAA-3′ (NUCKS1); S146, 5′- GAGAAUGGCAGGCUGUUUAUU-3′ (NEK5); S149, 5′-CUGAAU UCACGGAGCAAUU-3′ (CDK18/PCTK3); S150, 5′-CAACUGAAC CACCCAAAUA-3′ (NEK6); S151, 5′-GACCAGAACCGCUGGGAU U-3′ (CDKN1C); S155, 5′-UCAGAGCCACCGCAGAUUA-3′ (IRAK1); S157, 5′-CACAUUCGAAUCGGUAUAUUA-3′ (IRAK3); S158, 5′-AUG CAGGUUUGCUGGGUUU-3′ (N-acetylglucosamine kinase); S209,

5′-GGUCUUAUGCACAGAAGUA-3′ (EYA4); S211, 5′-GCAUUG AGAUUCCUGCAGA-3′ (PKM2); S212, 5′-GGACAGUGGCAGUGA AACA-3′ (polynucleotide kinase-phosphatase); S220, 5′-GAG AGGUGGUGGCGCUUAA-3′ (CDK2); S226, 5′-UUACAGAGGUUC AGGAUUA-3′ (ROS proto-oncogene 1); S227, 5′-GACAGGAGC ACCCUCAUUU-3′ (HK3); and S228, 5′-GGACAACAAUUUGCA UUAA-3′ (RIOK2).

## IF microscopy

Cells were seeded onto glass coverslips 1 d before drug treatment, fixed with 4% PFA for 10 min, permeabilized with 0.5% Triton X-100 in PBS for 10 min, and blocked with 3% BSA in PBS for 1 h at RT. Cy3- or Cy2-conjugated secondary donkey antibodies (Jackson ImmunoResearch Laboratories) were used for detection of primary antibodies. DNA was stained with Hoechst dye (1:10,000, Sigma-Aldrich). Coverslips were mounted onto glass slides using a solution of 14% polyvinyl-alcohol (P8136, Sigma-Aldrich) and 30% glycerol in PBS. Microscopy was performed on a Leica DM 5000 Microscope using a 20× (NA 0.70) or 40× dry objective (NA 0.75), or a 40× oil objective (NA 1.25–0.75). Alternatively, a Nikon Eclipse Ti2-E microscope was used in combination with a 20× (NA 0.95) or a 40× dry objective (NA 0.75) or a 40× oil objective (NA 1.4). Images were taken with an Andor charge-coupled device camera or a pco edge sCMOS camera, processed by adjusting the brightness and contrast, and analyzed using Adobe Photoshop and Fiji software.

## Western blot analysis

Cells were lysed by scraping in ice-cold protein lysis buffer (50 mM Tris-HCl, pH 7.4, 150 mM NaCl, 15 mM MgCl$_2$, and 1% Triton X-100) supplemented with EDTA-free protease inhibitor cocktail (Roche) and phosphatase inhibitors (1 mM sodium vanadate, 50 mM sodium fluoride, and 0.04 µM okadaic acid). Samples were incubated for 5 min on ice, and nuclei were removed by centrifugation for 5 min at 10,000 $g$ at 4°C. 10–20 µg total protein diluted in SDS sample buffer (4% SDS, 20% glycerol, 10% DTT, 0.004% bromophenol blue, and 0.125 M Tris HCl) was loaded onto 5–20% polyacrylamide gradient gels and transferred to a 0.2-µm pore size nitrocellulose membrane (PeqLab) by wet blotting. Membranes were blocked in 5% milk or 5% BSA (both diluted in PBS) at RT, incubated with primary antibodies diluted in PBS containing 0.1‰ sodium-azide overnight at 4°C and washed with TBS containing 1% Tween 20. HRP-conjugated secondary antibodies (Jackson Immunoresearch, diluted 1:5,000 in PBS) and Western Lightning Enhanced Chemiluminescence substrate (PerkinElmer) were used for detection.

## Antibodies and DNA oligonucleotide primers

The following antibodies were used: mouse anti-G3BP1 (TT-Y; Santa Cruz, sc-81940), mouse anti-acetylated tubulin (Sigma-Aldrich, C3B9), goat anti-eIF3B (Santa Cruz, sc-16377), mouse anti-puromycin (Millipore, MABE343), mouse anti-CDK1 (B-6; Santa Cruz, sc-8395), mouse-anti-CDK2 (Santa Cruz, sc-6248), rabbit anti-RPS6 (5G10; Cell Signaling, 2217), rabbit anti-phospho-eIF2α (S51; Cell Signaling, 9721), rabbit anti-eIF2α (Cell Signaling, 9722), rabbit anti-4EBP1 (Cell Signaling, 9644), rabbit anti-phospho-4EBP1 (T37/46; Cell Signaling, 236B4), rabbit

anti-phospho-RPS6 (S235/236; D57.2.2E; Cell Signaling, 4858), mouse anti-phospho-p70 S6K1 (T389; 1A5; Cell Signaling, 9206), rabbit anti-p70 S6K1 (49D7; Cell Signaling, 2708), rabbit anti-mTOR (7C10; Cell Signaling, 2983), mouse anti-tubulin (DM1A; Sigma-Aldrich, T9026), rabbit anti-cyclin A (H-432; Santa Cruz, sc-751), rabbit anti-cyclin B1 (Cell Signaling, 4138), rabbit anti-cyclin D1 (EPR2241; Abcam, ab134175), rabbit anti-Histone H3 (Abcam, ab1791), rabbit anti-phospho-H3 (S10; Abcam, ab5176), rabbit anti-eIF3A (D51F4; Cell Signaling, 3411), mouse anti-HA.11 (MMS-101P, Covance), rabbit anti-RPS10 (Abcam, ab151550), rabbit anti-RPL12 (Proteintech, 14536-1-AP), mouse anti-RPS3 (Santa Cruz, sc-376098), rabbit anti-LARP1 (Abcam, ab86359), mouse anti-FLAG M2 (Sigma-Aldrich, F 3165), mouse anti-eIF4E (P-2; Santa Cruz, sc-9976), rabbit anti-eIF4G (Santa Cruz, sc-11373), goat anti-eIF4AI (Santa Cruz, sc-14211), and mouse anti-GAPDH (Thermo Fisher Scientific, AM4300).

DNA oligonucleotide primers were as follows: DNA G1714, 5′-GAAGGCTCATGGCAAGAAGG-3′ (β globin forward); G1715, 5′-ATGATGAGACAGCACAATAACCAG-3′ (β globin reverse); G2943 5′-TGGAGACTCTCAGGGTCGAAA-3′ (CDKN1A forward); G2944 5′-GGCGTTTGGAGTGGTAGAAATC-3′ (CDKN1A reverse); G2979, 5′-TCGATGGGCGATCTATTTCCCTGT-3′ (NCL forward); G2980, 5′-TGTTGCACTGTAGGAGAGGTTGCT-3′ (NCL reverse); G3007, 5′-GAGTTCGAGTCCGGCATCT-3′ (RPS7 forward); G3008, 5′-CGACCACCACCAACTTCAA-3′ (RPS7 reverse); G4542: 5′-GATCCCCCGGGAATAACATCCACTTTGCCTTTCTC-3′; G4543: 5′-TTTCCCGGGTCATAGATTCATACGCAGGTGC-3′; G4737, 5′-TCTACAGAAAACATGCCCATTAAG-3′ (EIF2S1 forward); G4738, 5′-GCCATAGCTTGACTGAGGACA-3′ (EIF2S1 reverse); G4739, 5′-TCTACAACCCTGAAGTGCTTGAT-3′ (RPLP0 forward); G4740, 5′-CAATCTGCAGACAGACACTGG-3′ (RPLP0 reverse); G4753, 5′-GTAGGCCGTGCACAAAGA-3′ (PABPC4 forward); G4754, 5′-AATGTAGAGATTCACCCCCTGA-3′ (PABPC4 reverse); G4976, 5′-CTGGGTGAAGAATGGAAGGGTT-3′ (RPS6 forward); G4988, 5′-TGCATCCACAATGCAACCAC-3′ (RPS6 reverse); LARP1-oligo1, 5′-CACCGAGACACATACCTGCCAATCG-3′; LARP1-oligo2, 5′-AAACCGATTGGCAGGTATGTGTCTC-3′; LARP1-oligo3, 5′-GGGAAAGGGATCTGCCCAAG-3′; LARP1-oligo4, 5′-CACCAGCCCCATCACTCTTC-3′; 5′-TGGCCAAGTCCAAGAACCAC-3′ (RPL29 forward); 5′-CAAAGCGCATGTTCCTCAGG-3′ (RPL29 reverse); 5′-CCAGATCTTGATGCCCAACA-3′ (RPL32 forward); 5′-TTTGCGGTTCTTGGAGGAAA-3′ (RPL32 reverse); 5′-CCAGAGGCGTACAGGGATAG-3′ (ACTB forward); and 5′-TGGCACCACACCTTCTACAA-3′ (ACTB reverse).

## Polysome profile analysis

Cells were seeded 1 d before the experiment and kept at sub-confluence to prevent translation suppression by contact inhibition. Cells were then treated with 100 µg/ml cycloheximide (CHX) for 5 min at RT to stabilize existing polysomes before washing with ice-cold PBS and harvesting by scraping in polysome lysis buffer (20 mM Tris HCl, pH 7.5, 150 mM NaCl, 5 mM MgCl$_2$, 1 mM DTT, 100 mg/ml CHX, 1% Triton X-100, 40 U/ml RNasin, and EDTA-free complete protease inhibitors [Roche]). Lysates were rotated end-over-end for 10 min at 4°C and cleared by centrifugation at 10,000 $g$ for 10 min at 4°C. 40 µl lysate was saved for Western blot analysis before the cellular lysate was loaded onto linear 17.5–50% sucrose gradients (dissolved in

20 mM Tris-HCl, pH 7.5, 5 mM MgCl$_2$, and 150 mM NaCl). For polysome disruption, lysates were digested either with RNase I (60 units per OD260, Ambion) or with a combination of RNase A (71.4 ng per OD260, Thermo Fisher Scientific) and RNase T1 (42.9 units per absorbance at 260 nm [$A_{260}$], Thermo Fisher Scientific) for 5 min at 4°C. Sucrose density gradient centrifugation was performed at 35,000 rpm at 4°C using a SW60 rotor (Beckman) for 2.5 h. Polysome profiles were recorded by measuring the $A_{254}$ using a Teledyne ISCO Foxy Jr. or a Teledyne ISCO Foxy R1 system in combination with PeakTrak software. Profiles were aligned manually according to the 80S peak, and the percentage of polysomal ribosomes was calculated by dividing the area under the curve of the polysomal ribosomes by the total area under the curve.

## Polysome fractionation

During gradient elution, fractions of ∼300 µl were collected every 14 s. For RNA isolation, 300 µl urea buffer (10 mM Tris, pH 7.5, 350 mM NaCl, 10 mM EDTA, 1% SDS, and 7 M urea) containing 25 fmol rabbit HBB2 in vitro transcript and 300 µl phenol:chloroform:isamylalcohol (25:24:1) were added to each fraction. After phase separation, RNA was isolated from the aqueous phase and precipitated using isopropanol. RNA levels in the different fractions were subsequently analyzed by quantitative PCR (qPCR) as follows: RNA was reverse transcribed using Moloney murine leukemia virus reverse transcription (Promega), followed by cDNA amplification using the PowerUp SYBR Green Master Mix (Thermo Fisher Scientific) and the Quant-Studio 5 Real-TimePCR system (Thermo Fisher Scientific). All threshold count values were normalized to the HBB2 spike-in transcript to correct for isolation differences.

For protein purification, 300 µl Tris-HCl (20 mM, pH 7.5) and 10 µl StrataClear beads were added to each fraction. Samples were rotated end-over-end at 4°C overnight and centrifuged at ∼100 $g$ for 2 min, and proteins were eluted from the beads using SDS sample buffer.

## Ribo-Seq analysis

RPE1 cells were cultured in the absence of FBS for 48 h. Afterward, cells were incubated for 4 h in fresh medium without FBS supplemented with either DMSO or Ro3306 (10 µM), washed once in ice-cold PBS supplemented with 100 µg/ml CHX, and harvested by scraping in polysome lysis buffer. Lysates were rotated end-over-end for 10 min at 4°C and cleared by centrifugation at 10,000 $g$ for 10 min at 4°C. The DMSO- and Ro3306-treated samples were adjusted to the same $A_{260}$ before yeast polysome lysate (2% of the RPE1 lysates according to $A_{260}$ measurement) was spiked into each sample. 10% of the lysates were saved as input samples. The lysates were subsequently digested with RNase I (60 units per $A_{260}$; Ambion) for 5 min at 4°C, and the reaction was stopped by addition of Superase inhibitor (six units; Invitrogen). Samples were then fractionated by 17.5–50% sucrose density gradient centrifugation, and RNA was purified from the cytoplasmic lysate (input) or from the monosomal fractions (ribosome protected fragments) using phenol:chloroform:isamylalcohol (25:24:1) by phase separation. Both input and ribosome protected fragments were depleted of ribosomal RNA

(rRNA) with the Ribo-Zero Gold Kit (Illumina). Input RNA was randomly fragmented by alkaline hydrolysis at pH 10.0 for 12 min at 95°C. Fragmented RNA and ribosome protected fragments were size-selected (25–35 nt) on a 15% polyacrylamide Tris-borate-EDTA-urea gel. After end-repair with T4 PNK, 3 ng per sample was used for library preparation using the NEXTflex Small RNA-Seq Kit v3 according to the manufacturer's manual. Libraries were multiplexed and sequenced on one lane of a NextSeq500 sequencer (Illumina).

For Ribo-Seq data analysis, adapter sequences were first removed with the FASTX-toolkit (http://hannonlab.cshl.edu/fastx_toolkit/), and the four random nucleotides at the beginning and end of the reads were trimmed. Read alignment was then performed using Bowtie (Langmead et al., 2009). Reads that did not map to human tRNA or rRNA sequences were aligned to a common human transcriptome reference (wgEncodeGencodeBasicV27) and a yeast transcriptome (sacCer3-ensGene). To summarize reads at the gene level, only reads that mapped to the annotated ORF of isoforms of one specific gene (as defined by a common gene symbol) were counted with an in-house-developed Perl script. To identify individually regulated mRNAs with DESeq2, human read counts were normalized with the median ratio method before calculating mean fold changes, and P values for changes in RD were obtained from a likelihood ratio test (Love et al., 2014). The sum of read counts assigned to yeast or human ORFs was used to measure global changes in translation efficiency. For categorization, mRNAs that contained an IRES element (according to http://iresite.org/IRESite_web.php?page=browse_cellular _transcripts) or a 5′TOP motif (according to Meyuhas and Kahan [2015]) were grouped. A detailed method description and the processed data are deposited in the GEO database (accession no.: GSE128538).

### Puromycin incorporation

Cells were treated with 10 µg/ml puromycin (Gibco BRL Life Technologies) for 5 min at 37°C, washed twice with PBS, and lysed in protein lysis buffer for Western blot analysis or fixed in ice-cold methanol for 3 min for IF microscopy. For Western blot analysis, equal amounts of total cell lysates were separated by SDS-PAGE. Puromycin signals were detected with anti-puromycin antibody. The signal intensity was measured along the entire lane and normalized to the overall Ponceau S staining of the corresponding lane. For IF microscopy, cells were blocked with 3% BSA in PBS for 1 h at RT, followed by incubation with anti-puromycin and anti-eIF3B antibodies. IF staining and microscopy were performed as described above. Puromycin signal intensities were measured using Fiji software as follows. Cells were first segmented into regions of interest using the Hoechst and eIF3B signals to detect cells and cell borders. Afterward, mean intensities for every region of interest were measured and normalized to the average mean intensity of S14 nontargeting control siRNA transfected cells.

### Cell cycle analysis by FACS

Cells were collected by trypsinization and resuspended in PBS containing 1% FCS (PAA Laboratories). Cells were then pelleted at 780 $g$ for 3 min, resuspended in 200 µl PBS, and fixed with 800 µl ice-cold 70% ethanol. After 10 min, cells were resuspended in PBS containing 10 mg/ml PI (Sigma-Aldrich) and 0.5 mg/ml RNase A (Sigma-Aldrich). Cell cycle profiles were analyzed using a FACS Canto II flow cytometer (BD Biosciences).

### Phosphoproteomics

HeLa cells were cultured for 14 d in SILAC medium (DMEM without arginine, lysine, glutamine, and pyruvate, containing 10% FBS for SILAC [Silantes], 2 mM L-glutamine, 100 U/ml penicillin, 100 µg/ml streptomycin [all PAN Biotech], and either light- or heavy-labeled amino acids [SILAC amino acids, Silantes 211603902 and 201603902]). Heavy- and light-labeled cells were seeded into four 15-cm dishes each, $3.5 \times 10^6$ cells per dish. Light-labeled cells were treated with DMSO and heavy-labeled cells with Ro3306 (10 µM) for 4 h, and labels were swapped in the repeat experiment. Cells were washed with ice-cold PBS, 200 µl low-magnesium polysome lysis buffer (20 mM Tris HCl, pH 7.5, 150 mM NaCl, 0.25 mM MgCl₂, 1 mM DTT, 100 mg/ml CHX, 1% Triton X-100, 40 U/ml RNasin, EDTA-free complete protease inhibitors [Roche], and phosphatase inhibitors [PhosphoSTOP, Roche]) was added, and lysates were harvested by scraping. Lysates were then rotated end-over-end for 10 min at 4°C and cleared by centrifugation at 10,000 $g$ for 10 min at 4°C. The total protein content of the lysate was measured using a Bradford assay. Equal amounts of total protein from the heavy and the light sample were mixed and loaded onto low-magnesium 17.5–50% sucrose gradients (dissolved in 20 mM Tris-HCl, pH 7.5, 5 mM MgCl₂, and 150 mM NaCl). Sucrose density gradient centrifugation was performed as described above, and ribosomal fractions were pooled. Proteins were precipitated using the Wessel–Flügge precipitation protocol (Wessel and Flügge, 1984). Samples were enriched for phosphopeptides using PhosSelect iron affinity gel IMAC beads (first repeat) or TiO₂-SIMAC-HILIC (TiSH) phosphopeptide enrichment and fractionation (second repeat), and subsequently subjected to mass spectrometry and Maxquant analysis at the Core Facility for Mass Spectrometry & Proteomics of the Center for Molecular Biology of Heidelberg University. Gene ontology annotations were added using Perseus software.

### RNA-IP

For RNA-IP, endogenous LARP1 was immunoprecipitated from HeLa cell lysates using anti-LARP1 antibody. Briefly, cells were grown in 10-cm plates to 80% confluence and treated with either vehicle (DMSO) or Ro3306 (Sigma-Aldrich) at 10 µM for 4 h. Cells were washed in cold PBS and lysed for 5 min on ice in 900 µl of ice-cold hypotonic lysis buffer (10 mM Tris-HCl, pH 7.4, 10 mM NaCl, 2 mM EDTA, 0.1% Triton X-100, and Complete Mini EDTA-free protease inhibitor [Roche]). The NaCl concentration was subsequently readjusted to 150 mM. Lysates were cleared by centrifugation at 20,000 $g$, and 800 µl of supernatant was loaded onto protein G beads (20 µl bead volume; Sigma-Aldrich) that were precoupled and equilibrated with either anti-LARP1 or control IgG for 2 h at 4°C. After eight washes in NET-2 buffer (50 mM Tris-HCl, pH 7.4, 150 mM NaCl, and 0.1% Triton X-100), samples were eluted either by addition of 2× SDS sample buffer (10% mercaptoethanol, 100 mM Tris-HCl, pH 6.8, 2% SDS,

and 0.02% bromophenol blue) and heated for 3 min at 90°C (for Western blot) or by addition of 1 ml Trizol (Thermo Fisher Scientific) to the magnetic beads (for RNA isolation). RNA was isolated from the Trizol fraction according to the manufacturer's protocol. Input and immunoprecipitated RNA was quantified by qRT-PCR, amplifying RPL29, RPL32, and ACTB mRNAs. Briefly, samples were treated with DNase I (Thermo Fisher Scientific), and cDNA synthesis was performed using the Maxima First Strand cDNA synthesis kit for qPCR (Thermo Fisher Scientific) according to manufacturer's protocol. qPCR was performed using gene-specific primers and Platinum SYBR Green qPCR Supermix-UDG (Thermo Fisher Scientific) in an AriaMx real-time PCR apparatus (Agilent Technologies).

### Cap pulldown assay

Cells were lysed by scraping in cap pulldown lysis buffer (50 mM Tris-HCl, pH 7.8, 150 mM NaCl, 1 mM EDTA, 0.5% NP-40, and EDTA-free complete protease inhibitors [Roche]). Lysates were then rotated end-over-end for 10 min at 4°C, and nuclei were pelleted by centrifugation at 10,000 $g$ for 10 min at 4°C. 10% of total cell lysates were saved as input samples. 50 µl of γ-aminophenyl-m$^7$GTP agarose C10-linked beads (Jena Biosciences) was added to the remaining sample, which was then rotated end-over-end at 4°C overnight. Beads were washed five times using cap pulldown wash buffer (10 mM Tris-HCl, pH 7.8, 150 mM NaCl, 1 mM EDTA, and 0.1% NP-40) and eluted with SDS sample buffer.

### Statistical analysis

Statistical analysis was performed using Microsoft Excel 2010 or Prism software (GraphPad). Statistical significance was calculated by performing a one-tailed, paired Student's $t$ test whenever an equal number of repeats was performed for every condition. For unequal numbers of repeats, an unpaired $t$ test with unequal variance (Welch's $t$ test) was performed. When all values were calculated relative to control treatments, control samples were set to 1, all values were log-transformed, and a one-tailed ratio paired $t$ test (when comparing to a control value) or a one-tailed paired Student's $t$ test (when comparing two or more fold changes) was performed.

### Online supplemental material

Fig. S1 shows cell morphology of HT1080 and HT2-19 cells upon down-regulation of CDK1, cell cycle analyses upon Ro3306 treatment, HeLa-FUCCI cell distributions, and cilia formation in RPE1 cells in response to serum starvation. Fig. S2 shows polysome profiles and raw values for polysomal ribosome percentages that were used to calculate the fold changes in polysomal ribosomes depicted in Fig. 5 and Fig. 6. Fig. S3 shows the association of CDK1 with polysomes and the two individual phosphoproteomics experiments whose overlap is depicted in Fig. 7 B. Fig. S4 provides the quality control assessment for the ribosome footprint experiment depicted in Fig. 8 A. Fig. S5 provides repeat experiments of polysome fractionation experiments in HeLa and HEK293T cells, polysome profiles and their quantification recorded from HEK293T WT and LARP$^{-/-}$ cells, and a LARP1 RNA-IP experiment.

## Acknowledgments

We thank Andy Porter (Imperial College Faculty of Medicine, London, UK) for generously providing the HT2-19 cells, Randal Kaufman (Sanford Burnham Prebys Medical Discovery Institute, La Jolla, CA, USA) for the eIF2α-AA/SS MEFs, Nahum Sonenberg (McGill University, Montreal, Canada) for the 4EBP1/2$^{-/-}$ MEFs, Oded Meyuhas (Hebrew University of Jerusalem, Jerusalem, Israel) for the RPS6$^{P+/+}$ and RPS6$^{P-/-}$ MEFs, and Ingrid Hoffmann as well as Francesca Tuorto (both German Cancer Research Center, Heidelberg, Germany) for RPE1 cells and primary MEFs, respectively. We also thank John Blenis (Weill Cornell Medicine, New York, NY) for the S6K1 plasmids, Sarah Blagden (University of Oxford, Oxford, UK) and Bruno Fonseca (University of Ottawa, Ottawa, Canada) for LARP1 plasmids, James Hsieh (Cornell University, New York, NY) for the pTRE-Tight mTOR plasmids, Jamal Tazi (Institut de Génétique Moléculaire de Montpellier, Montpellier, France) for the G3BP1 cDNA, Atsushi Miyawaki (RIKEN Center for Brain Science, Japan) for the FUCCI plasmids, and Thomas Mayo (Harvard Medical School, Boston, MA) for help with cloning GFP-G3BP1. We are thankful to Ulrike Friedrich, Guenter Kramer, and Bernd Bukau (all at the Center for Molecular Biology of Heidelberg University, Germany) for providing yeast lysates and assistance with the Ribo-Seq protocol. Mass spectrometry analysis was carried out by the Core Facility for Mass Spectrometry & Proteomics at the Center for Molecular Biology of Heidelberg University (ZMBH). FACS sorting and analysis was done in the FACS core facility at the ZMBH and the Flowcore Mannheim of Heidelberg University. siRNA library preparation was performed by the Cellnetworks Advanced Biological Screening Facility of Heidelberg University, and next-generation sequencing was carried out by the Cellnetworks Deep Sequencing Core Facility of Heidelberg University.

This work was funded by the Deutsche Forschungsgemeinschaft, project number 201348542-SFB 1036 to G. Stoecklin and M. Knop, project number 278001972-TRR 186 to A. Ruggieri and G. Stoecklin, and grant INST 35/1067-1 FUGG to M. Knop; the Nationales Centrum für Tumorerkrankungen, project number NCT3.0 Integrative Project in Cancer Research NCT3.0_2015.54 DysregPT to G. Stoecklin; and the Danish Council for Independent Research–Natural Sciences, grant 6108-00197B to C.K. Damgaard.

The authors declare no competing financial interests.

Author contributions: K. Haneke performed the screen and carried out most experiments; J. Schott did the Ribo-Seq bioinformatics analysis; D. Lindner prepared the Ribo-Seq libraries; C. Mongis and M. Knop were instrumental for the imaging screen; A. Ruggieri generated HeLa-FUCCI cells; W. Palm generated the mTOR MEFs; A.K. Hollensen and C.K. Damgaard generated the LARP1$^{-/-}$ HEK293T cells and performed the RNA-IP; K. Haneke and G. Stoecklin designed the study, analyzed the data, and wrote the manuscript; and all authors approved the manuscript.

Submitted: 21 June 2019

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

# Supplemental material

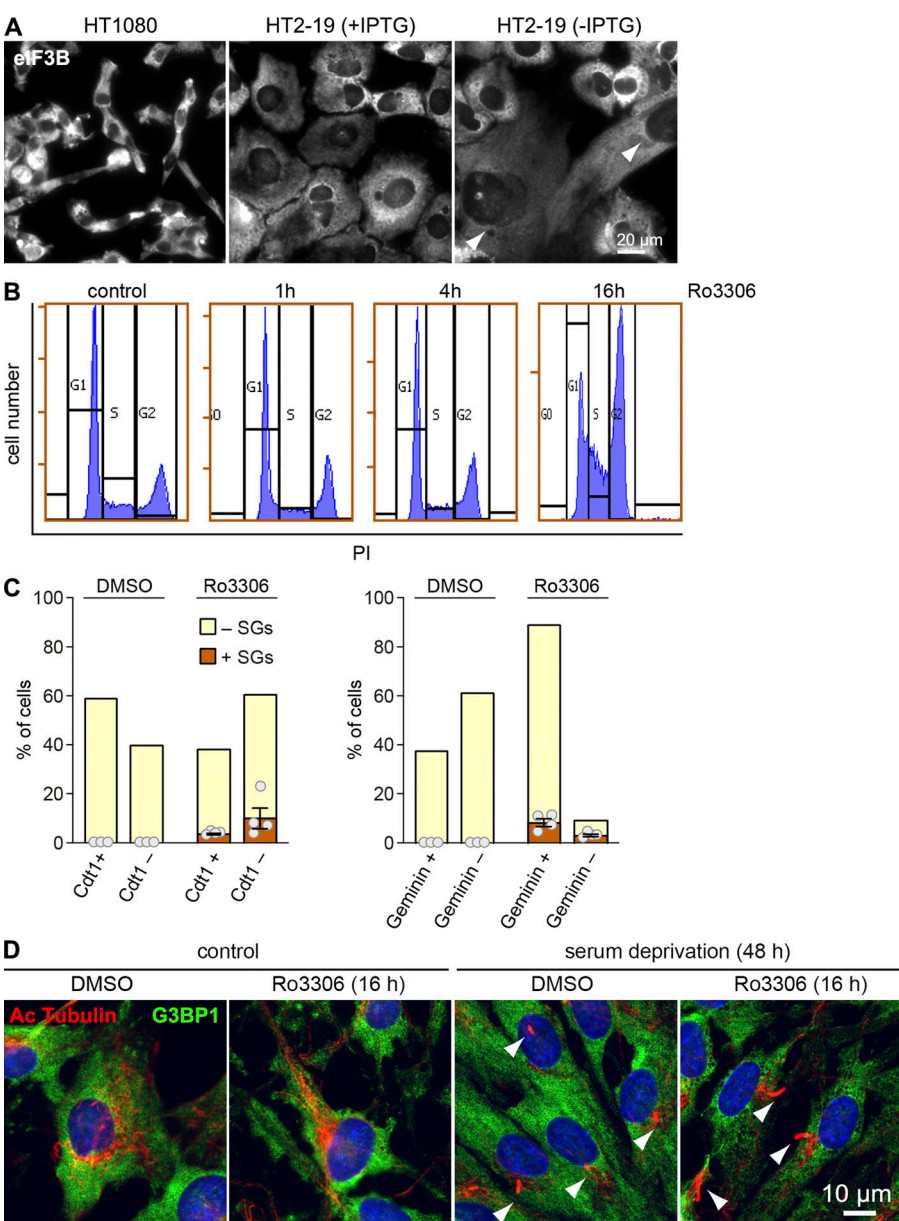

Figure S1.    **Pharmacological inhibition and genetic depletion of CDK1. (A)** HT1080 and HT2-19 cells were cultured in the presence or absence of IPTG (0.2 mM) for 7 d. Cellular morphology was assessed by IF microscopy of fixed cells stained with anti-eIF3B antibody. Arrows indicate the increase in cell size; scale bar = 20 μm. **(B)** HeLa cells were treated with control solvent (DMSO) or Ro3306 for 1, 4, and 16 h. Cell cycle profiles were recorded by FACS using PI staining. **(C)** Nonnormalized values of the SG quantification in HeLa FUCCI cells are shown in Fig. 4 C. **(D)** RPE-1 cells were serum-deprived for 48 h and subsequently treated with DMSO or Ro3306 (10 μM) for 16 h. Formation of cilia was analyzed by IF microscopy of fixed cells stained with anti-acetylated tubulin and anti-G3BP1 antibody. Arrows point toward cilia; scale bar = 10 μm.

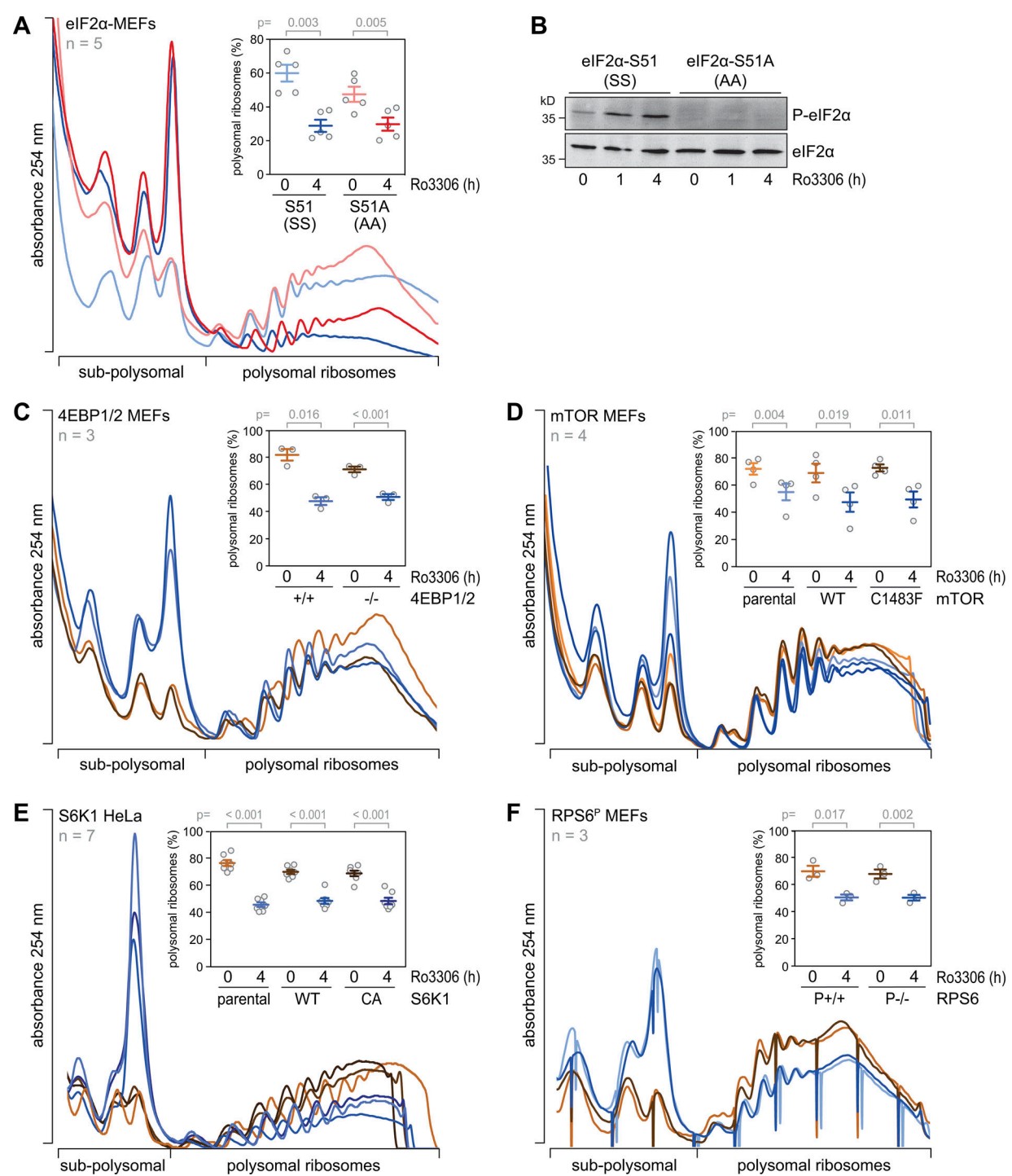

Figure S2. **Signaling pathways downstream of CDK1i. (A)** SS and AA MEFs were treated with DMSO or Ro3306 (10 µM) for 4 h. Polysome profiles were recorded after sucrose density gradient centrifugation; the percentage of polysomal ribosomes is represented in the inset (mean ± SEM, $n$ = 5). **(B)** The phosphorylation level of eIF2α (S51) was assessed in cytoplasmic lysates from samples in A by Western blot analysis. **(C–F)** Polysome profile analyses were quantified as in A from 4EBP1/2^{+/+} and 4EBP1/2^{−/−} MEFs (C); SV40-immortalized K-Ras^{G12D} MEFs (parental), and corresponding MEFs overexpressing mTOR-WT or the hyperactive mTOR mutant C1483F (D); parental, HA-S6K1-WT or HA-S6K1-CA expressing HeLa cells (E); and RPS6 WT (RPS6^{P+/+}) and RPS6 phosphodeficient S235A, S236A, S240A, S244A, and S247A (RPS6^{P−/−}) MEFs treated with either DMSO or Ro3306 (10 µM) for 4 h (F); the percentage of polysomal ribosomes is represented in the insets. Statistical significance in A and C–F was determined by paired one-tailed Student's $t$ test.

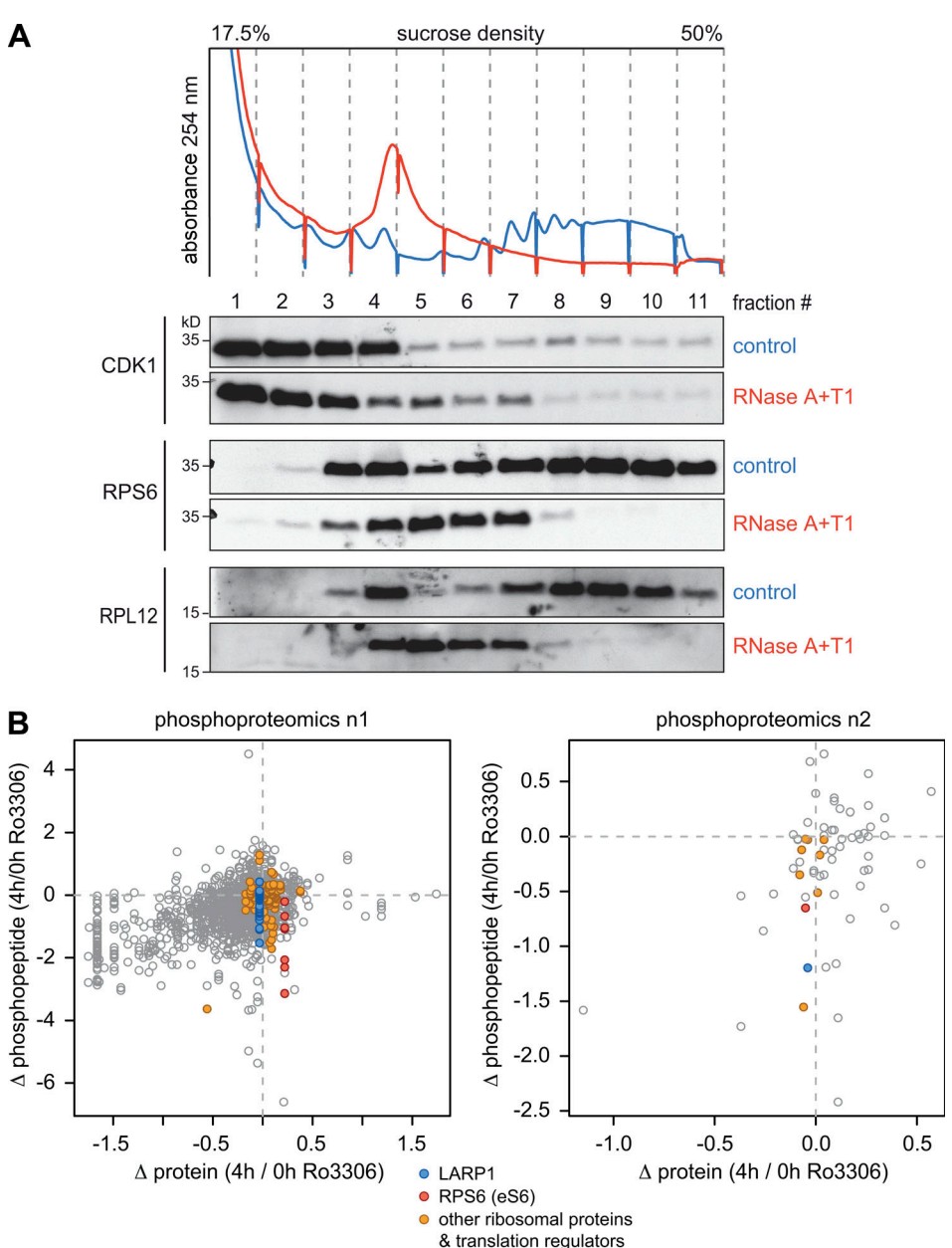

Figure S3. **Ribosome-associated CDK1-dependent phosphorylation events. (A)** HeLa cell lysates, either untreated or subjected to RNase A/T1 digestion, were fractionated following sucrose density gradient centrifugation. Association of CDK1, RPS6, and RPL12 with the different fractions was monitored by Western blot analysis. **(B)** For phosphoproteomics of ribosomal fractions, HeLa cells were SILAC labeled and treated with either DMSO (n1 light, n2 heavy) or Ro3306 (n1 heavy, n2 light) for 4 h. After lysis and disassembly of polysomes in low-magnesium buffer, samples were mixed, and ribosomal fractions obtained by sucrose density centrifugation were subjected to Wessel–Flügge precipitation. Phosphopeptides were enriched using either PhosSelect iron affinity gel IMAC beads (left panel, n1) or $TiO_2$ beads (right panel, n2), fractionated, and analyzed by mass spectrometry followed by MaxQuant analysis. For all phosphopeptides detected under both conditions, the ratio ($\Delta$ phosphopeptide abundance, 4 h/0 h Ro3306) was plotted against the ratio of the corresponding total protein ($\Delta$ protein abundance, 4 h/0 h Ro3306). Phosphopeptides derived from LARP1 (blue), RPS6 (red), and other translation regulators (orange) are color-coded.

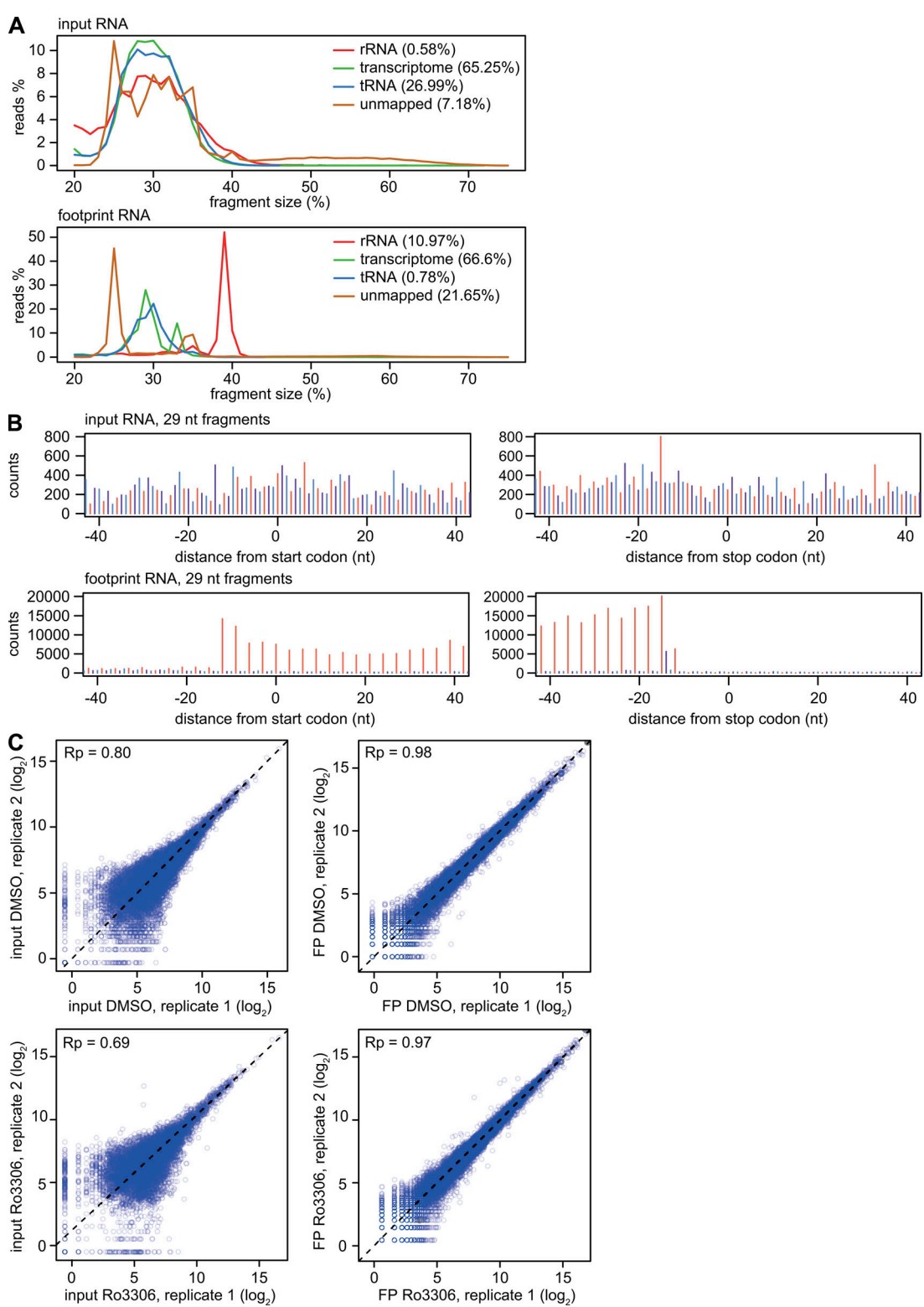

Figure S4. **Ribo-Seq quality assessment. (A)** All reads from the input RNA and ribosome footprint samples were mapped to human tRNA sequences, rRNA sequences, the human transcriptome (wgEncodeGencodeBasicV27), and a yeast transcriptome (sacCer3ensGene). Depicted is the fragment size distribution of the four groups (rRNA, tRNA, transcriptome, and unmapped) from one representative input RNA (top) and footprint sample (bottom). **(B)** Reads from one representative input RNA (top) and footprint sample (bottom) were aligned at the 5' end, and the distribution of all 29-nt-long fragments is depicted according to their position relative to the start codon (left) and stop codon (right). **(C)** To assess reproducibility between repeat experiments, two repeats are plotted against each other representative for each condition (input DMSO, input Ro3306, footprint DMSO, and footprint Ro3306).

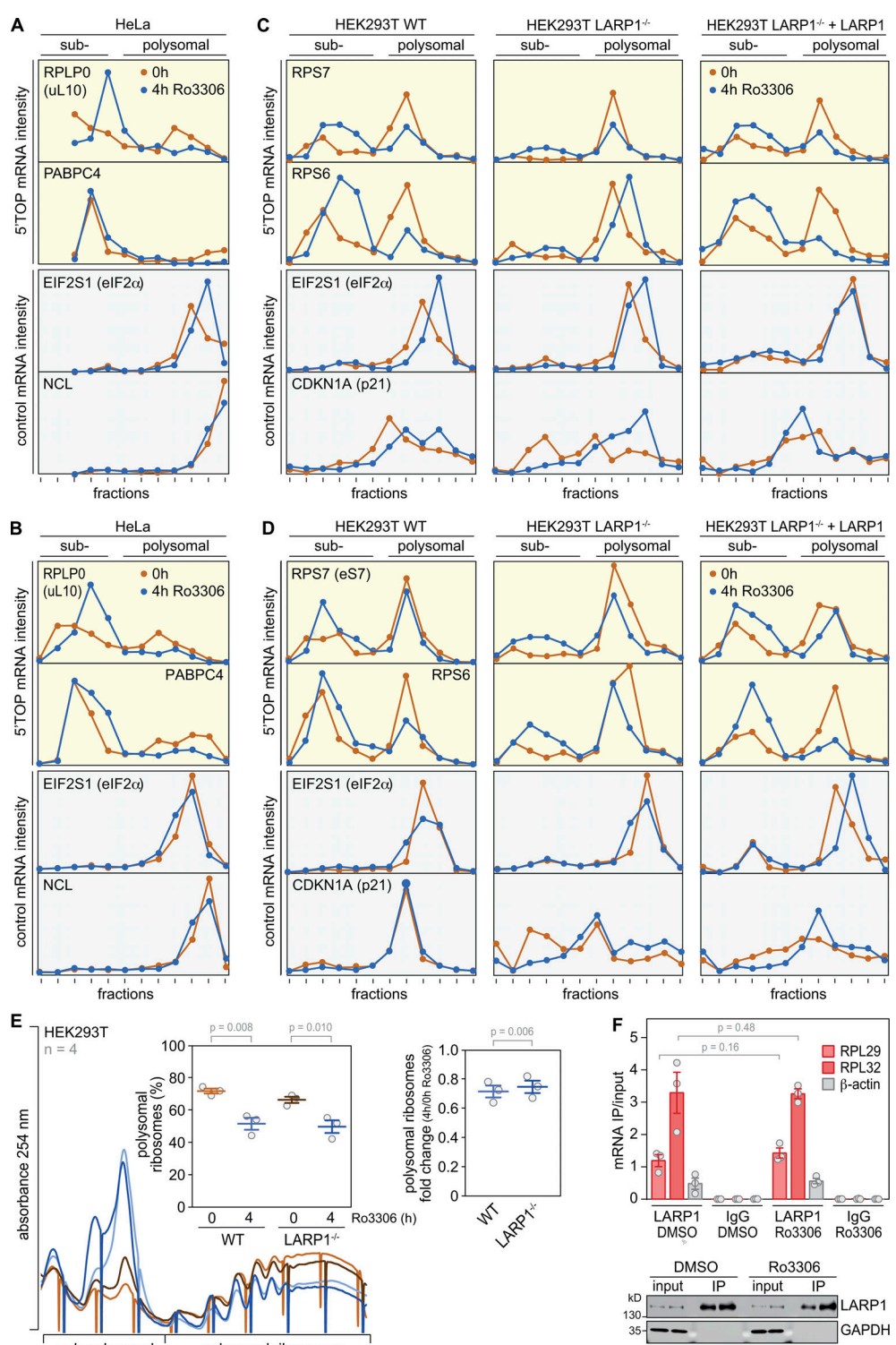

**Figure S5. LARP1 in suppression of 5'TOP mRNA translation by CDK1i. (A and B)** Polysome association of 5'TOP (RPLP0 and PABPC4) and ORF size-matched non-TOP (EIF2S1 and NCL) mRNAs was analyzed by polysome fractionation and subsequent qPCR analysis from DMSO- or Ro3306-treated (10 µM, 4 h) HeLa cells. **(C and D)** Polysome association of 5'TOP (RPS6 and RPS7) and ORF size-matched non-TOP (EIF2S1, CDKN1A) mRNAs was analyzed by polysome fractionation and subsequent qPCR analysis from DMSO- or Ro3306-treated (10 µM, 4 h) HEK293T WT, LARP1$^{-/-}$, and LARP1$^{-/-}$ + LARP1 cells. **(E)** Polysome profiles were recorded from HEK293T WT and LARP1$^{-/-}$ cells treated with either DMSO or Ro3306 (10 µM) for 4 h. The percentage of polysomal ribosomes is represented in the inset (mean ± SEM, n = 3), and the graph on the right side depicts the fold change in polysomal ribosomes (4 h Ro3306/DMSO control). **(F)** HeLa cells were treated with DMSO or Ro3306 (10 µM) for 4 h, and cell lysates were prepared for RNA-IP using anti-LARP1 or IgG control antibody. The amount of LARP1- and IgG-bound 5'TOP (RPL29 and RPL32) and non-TOP β-actin mRNAs was determined by qPCR (mean ± SEM, n = 3). Protein levels of LARP1 and GAPDH (negative control) in input and IP samples were measured by Western blot analysis. In E and F, statistical significance was determined by paired one-tailed Student's t test.

