## [Peer Review File · The Journal of Cell Biology]

CDK1 couples proliferation with protein synthesis

Katharina Haneke, Johanna Schott, Doris Lindner, Anne Hollensen, Christian Damgaard, Cyril Mongis, Michael Knop, Wilhelm Palm, Alessia Ruggieri, and Georg Stoecklin

Corresponding Author(s): Georg Stoecklin, Medical Faculty Mannheim of Heidelberg University

Review Timeline:

Submission Date:	2019-06-21
Editorial Decision:	2019-08-04
Revision Received:	2019-11-20
Editorial Decision:	2019-12-12
Revision Received:	2020-01-07

Monitoring Editor: Ian Macara

Scientific Editor: Marie Anne O'Donnell

Transaction Report:

DOI: <https://doi.org/10.1083/jcb.201906147>

August 4, 2019

Re: JCB manuscript #201906147

Prof. Georg Stoecklin
Medical Faculty Mannheim of Heidelberg University
Biochemistry
Ludolf-Krehl-Strasse 13-17
CBTM
Mannheim 6968167
Germany

Dear Georg,

We have now received reviews on your manuscript "CDK1 couples proliferation with protein synthesis" from two external referees with expertise in the field. I am pleased to say that both found the work to be of interest. However, they also both raised a number of substantive issues that will need to be addressed before the Journal could consider moving forward with the manuscript.

Reviewer #1 points out that stress granule formation is a very indirect readout for a screen for regulators of global protein synthesis, and that a number of other candidates were found in addition to CDK1, so we feel that it would be important to discuss more clearly the rationale for choosing to focus on CDK1. In addition, the reviewer asks whether the effects of CDK1 inhibition on SG formation is directly linked to inhibition of translation, and suggests that siRNA of CDK1 would be an important alternative to small molecule inhibition. There are a number of other problems that are raised in this critique. The second reviewer also notes that the proposal that CDK1 does not act via mTOR is not convincing, and the data supporting the idea that CDK1 interacts directly with ribosomes is also rather weak. Again, this reviewer has numerous additional comments about specific figures, statistics, and experimental design. Overall, it appears that significant experimental work will be required to address these points, and for this reason you might elect to submit or transfer the manuscript elsewhere. If you decide to resubmit, we would need a point-by-point response to each of the reviewer comments. Please note that a revised manuscript would be sent for evaluation by the same external reviewers.

GENERAL GUIDELINES:

Text limits: Character count for an Article is < 40,000, not including spaces. Count includes title page, abstract, introduction, results, discussion, acknowledgments, and figure legends. Count does not include materials and methods, references, tables, or supplemental legends.

Figures: Articles may have up to 10 main text figures. Figures must be prepared according to the policies outlined in our Instructions to Authors, under Data Presentation,

<http://jcb.rupress.org/site/misc/ifora.xhtml>. All figures in accepted manuscripts will be screened prior to publication.

Supplemental information: There are strict limits on the allowable amount of supplemental data. Articles may have up to 5 supplemental figures. Up to 10 supplemental videos or flash animations are allowed. A summary of all supplemental material should appear at the end of the Materials and methods section.

The typical timeframe for revisions is three months; if submitted within this timeframe, novelty will not be reassessed at the final decision. Please note that papers are generally considered through only one revision cycle, so any revised manuscript will likely be either accepted or rejected.

Thank you for this interesting contribution to Journal of Cell Biology. You can contact us at the journal office with any questions, cellbio@rockefeller.edu or call (212) 327-8588.

Sincerely,

Ian Macara, Ph.D.
Editor

Marie Anne O'Donnell, Ph.D.
Scientific Editor

Journal of Cell Biology

Reviewer #1 (Comments to the Authors (Required)):

In this manuscript, Haneke et al. performed a siRNA screen to identify kinases and phosphatases that play a role in global protein synthesis. They used stress granule formation as a read-out of the screen and only focused on one target, CDK1. They concluded that CDK1 regulates translation independently from its role in cell cycle. The authors also provide some evidence that eIF2 α and S6K1 signaling pathways act downstream of CDK1 in controlling global protein synthesis. Additionally, they also propose that CDK1 phosphorylates LARP1, which in turn is required for the translation of 5'TOP mRNAs. Although this manuscript highlights the role of CDK1 in translational control independently of its function in cell cycle, this reviewer believes that there are some major concerns the authors should address before publication.

Major Concerns:

- The manuscript is based on a siRNA screen, which is designed to identify phosphatases and kinases as regulators of global protein synthesis. Although it is odd, the authors chose stress granule (SG) formation as a read-out of this screen. Induction of SG has been shown to be associated with different cellular stress and, therefore, can be a very indirect way to measure the translation efficiency in cells. In addition, they identified several kinases and a few phosphatases, but only focused on the CDK1 protein. Did the authors validate any other hits from the screen? As NAGK, HK3, ROS1, and several others have a higher SG score than CDK1, we can assume that they should have a larger effect on translation.
- The authors should address whether the role of CDK1 inhibition on SG formation is directly associated with the alterations in translational control.
- After a siRNA-based screen in HeLa cells, the authors performed all of the mechanistic experiments in the presence of a CDK1 inhibitor. It is unclear why the authors do not use a siRNA-based method to investigate the role of CDK1 in translation. For example, some of the key experiments, especially the polysome profiling and puromycin incorporation, should be repeated in HeLa cells expressing CDK1 siRNA/shRNA.
- The experiments delineated in Fig. 2E are not very well controlled. Instead of using 2 different cell lines, it would be better to compare one cell line expressing different CDK1 levels. Also, the authors should assess whether the CDK1 activity on global protein synthesis is restricted to cancer cell lines or if it is also important in normal cells.
- In Fig. 4, the authors show that CDK1 inhibitor promotes eIF2 α phosphorylation. For this experiment, it is crucial to recapitulate the increase of p-eIF2 α employing CDK1 KD cells. Is phosphorylation of eIF2 α observed in the phosphoproteomic data?
- Based on the phosphoproteomic analysis, CDK1 should phosphorylate the LARP1 protein. Can the authors validate this result by using western blot analysis? If so, instead of using LARP1^{-/-} HEK293T cells, the authors should use a phospho-deficient mutant of LARP1 in the presence, or absence, of Ro3306.

Minor Concerns

- The association of CDK1 with the ribosomes was tested by checking the presence of RPS10 in polysomal fractions. What about RPL12?
- The quality of the figures was very poor. Please provide figures with high resolution.

Reviewer #2 (Comments to the Authors (Required)):

The initial purpose of this study was to identify novel regulators of protein synthesis, by siRNA screen against 711 and 256 human kinases and phosphatases, respectively, using HeLa cells stably expressing GFP-G3BP1 which allow to use stress granule formation as a visual read-out. This elegant screen revealed an interesting list of candidates and the authors decided to further pursue CDK1. The study mainly claims that CDK1 enhances global protein synthesis in a cell cycle phase-independent manner to adjust protein synthesis to overall proliferation rate, rather than to a specific phase of the cell cycle. CDK1 was shown before to control protein synthesis mostly during mitosis, but was also suggested to serve as a more global regulator of protein synthesis due to its ability to phosphorylate 4E-BP1 (Heesom et al., 2001; Shuda et al., 2015). Therefore, the claim that CDK1 is a regulator of global synthesis outside of its role as a cell cycle regulator is not fully novel. However, the current manuscript aims to support this claim with by additional strong experiments but the problem is that unfortunately some of these experiments are not fully convincing as presented (see specific comments to Figure 3).

The claim that CDK1 adjusts the overall proliferation rate of the cell to the overall protein synthesis rate is important and interesting. However, first, it is not fully supported by the experiments presented in the current manuscript; second, a recent study by Miettinen T. et al., eLife 2019 titled "Mammalian cell growth dynamics in mitosis" shows that CDK1 leads to increased phosphorylation of 4E-BP1 and cap-dependent protein synthesis during a specific time window in mitosis and convincingly demonstrates the coupling of CDK1-dependent translation regulation to cell proliferation. This important publication, which was probably published after Haneke et al had submitted their manuscript, must be cited and appropriately discussed.

The important question the study addresses is how CDK1 elicits its job as a global regulator of protein synthesis. It is stated that "Cdk1 most likely does not act via mTOR" (page 11, line 325-6) which may indeed be the case but this is not convincingly supported experimentally. The authors favor the idea that CDK1 interacts directly with ribosomes, and directly phosphorylates additional ribosomes-associated proteins, including LARP1; but a strong evidence for this notion is lacking (see comments to Figure 5). The strongest section in the manuscript is the Ribo-Seq analysis and related experiments (Figs 6, S6, S7), which uncovered that CDK1 exerts a particularly strong effect on the translation of 5'TOP mRNAs. The effect of CDK1 on 5'TOP mRNA was convincingly demonstrated to be LARP1-dependent. The suggestion is that LARP1 phosphorylation itself is dependent on CDK1 activity, but this specific point remains open. Overall, this is an important and interesting study with partial novelty, which suffers from major problems as detailed below.

Comments:

(*) General comment about Statistics:

When the same control sample(s) are used in an experiments addressing more than one variable (condition), there is a requirement to either use Anova test (for example for Figs 2D, 4B,D) or to use student t-test with Benferroni correction. In addition, a paired Student's t-test should be used (unless there is a justified excuse to use unpaired t-test. Given the above, the statistics of many experiments presented in the manuscript must be re-calculated followed by changing some of the conclusions. For example, correct analysis of Fig. 4 reveals that phosphorylation level of eIF2a upon 4h treatment with Ro33606, is not significant ($P=0.08$).

General comment about graphs presentation

Instead of showing only the fold effects, it is important to present the entire data, including that of the untreated (or DMSO-treated) cells.

Specific comments

Figure 3

- % SG in untreated cells is not shown (Fig 3A). It might be 10% even without Ro3306 treatment.
- Fig 3A shows one field for each cell type. It is not clear if the cells were synchronized for this experiment or not. In the left image most of the cells are in G1 and one cell contains SG. In the right image most of the cells are in S/G2/M, while >3 cells contain SG, some of them seem to be at late stages of mitosis.
- Cells at early mitosis (just upon nuclear membrane break down) must be presented in the chosen field. They are expected to express mVenus-Geminin throughout the cell, which will combine with the color of eIF3B detector. It is important to present these cells in order to address the question related to SG status during mitosis.
- The statistics of this experiment is not clear (Fig. 3B: N=? and see also general comment about statistics *)
- The text at the bottom of page 6 does not match Fig. 3C and its legend. There is a confusion between arrest at G1 or S phase, and only 2 out of the 3 profiles are shown.

Figure 4

- Fig 4B-C- D-E: see the general comment related to statistics (*)

- Fig 4A: Quantification of some of the bands is problematic: (i) some are fused between lanes. It may be due to over loading and/or overexposure. Is the quantification within the linear range? (ii) it is difficult to understand which of the different P-4E-BP1 bands represent the phosphorylated forms and it is difficult to separate them for quantification; (iii) total eIF2a blot suffer from a strong background.

- The effect of Ro3306 after 24h is presented in Fig4A and mentioned in the main text, while statistical calculations are missing for this time point in Fig. 4B.

- 4E-BP1 phosphorylation decreases upon CDKi, not increases as indicated in page 7 line 186.

Figure 5

Fig 5A: the data is not convincing and very confusing.

- It is important to show the polysomal fractions that were used in the main Figure rather than in a Supp Fig.

- Although RNase treatment is expected to cause polysomes disassembly, polysomes are still mostly stable, based on the presence of RPS10 in fractions 7-8-9-10.

- Fractions of each of the untreated or RNase treated experiments, should be analyzed for RPS10 and CDK1 using different positions of the same lanes (following SDS-PAGE and blotting). However, according to the different pattern of the bands on the blots, it looks like RPS10 and CDK1 were detected using different gels.

- Why there is no RPS10 signal specifically in fraction 2 in both untreated and RNase treated experiments? Is there a biological explanation?

- Page 8, line 214-217: The strength of the statement related to association of CDK1 with polysomes does not fit the strength of the data shown in Fig5A.

Fig 5B:

- It is not mentioned how many replicates were used for MS detection. Moreover, there is no information about MS data analysis (FDR %, normalization, statistics, etc.). Therefore it is impossible to judge the strength of the statement in page 8, line 227 "A prominent reduction of phosphorylation upon CDKi was observed for RPS6 and LARP1". Some of the RPS6 and LRRP1 phosphopeptides are located around the zero number of the 5B graph while an explanatory comment about the meaning of this observation is absent. Moreover, the authors do not comment about RPS6 protein abundance which seems to slightly increase in response to CDK1i. In case this is due to statistics considerations (which are not mentioned), maybe some of the changes in phosphopeptides abundance should be ignored too, based on similar statistics considerations.

- Fig. S5B should be part of the main figure and an additional test should be added to explain the differences in location of the indicated phosphoproteins on the graphs.

Minor comments:

- Page 6 lines 136-138: To enhance clarity the following sentence should be changed to: "CDK1 levels were reduced at least 2-fold in HT2-19 cells cultured in presence of IPTG compared to the parental HT-1080 cells, and polysomal ribosomes decreased from 43% to 32%"

- Fig 2BCD: multiple treatment times with Ro3306 not indicated in the legend

- Fig S1: Add to the legend what the arrows indicate (also add the appropriate information to the text in page 6 line 140)

- Fig S2: Add to the legend what the arrows indicate

- Fig S3A: The graph is confusing. SS and AA cells should be presented in two separated graphs because they are not compared to each other (as Anova test was not performed). Note that this comment also applies to additional Figures throughout the manuscript.

Response to Reviewers:

Reviewer #1:

In this manuscript, Haneke et al. performed a siRNA screen to identify kinases and phosphatases that play a role in global protein synthesis. They used stress granule formation as a read-out of the screen and only focused on one target, CDK1. They concluded that CDK1 regulates translation independently from its role in cell cycle. The authors also provide some evidence that eIF2 α and S6K1 signaling pathways act downstream of CDK1 in controlling global protein synthesis. Additionally, they also propose that CDK1 phosphorylates LARP1, which in turn is required for the translation of 5'TOP mRNAs. Although this manuscript highlights the role of CDK1 in translational control independently of its function in cell cycle, this reviewer believes that there are some major concerns the authors should address before publication.

Major Concerns:

The manuscript is based on a siRNA screen, which is designed to identify phosphatases and kinases as regulators of global protein synthesis. Although it is odd, the authors chose stress granule (SG) formation as a read-out of this screen. Induction of SG has been shown to be associated with different cellular stress and, therefore, can be a very indirect way to measure the translation efficiency in cells.

Response: We agree with the reviewer that SG formation is an indirect read-out for translation suppression and that many types of stress induce translation inhibition and SG formation. Thus, our SG screen will also identify indirect effects on translation induced by a stress response. However, induction of a stress response is forcibly a general problem of any screen reporting on the global translation rate and does not depend on the read-out for translation. To assess whether effects are indirect, one has to explore the underlying signaling pathways in careful follow-up experiments, which we did for CDK1. In fact, our SG screen could also report on candidates that influence SG assembly or disassembly independently of controlling translation. Our validation experiments explored this extensively by performing polysome profile analysis and puromycin incorporation assays upon knockdown (KD), pharmacological inhibition and genetic inactivation of CDK1.

We chose to perform a SG screen because SGs are an easy microscopic read-out and are not strongly affected by variations in cell density. We experienced that puromycin incorporation is strongly affected by the number and density of cells, which is difficult to control perfectly in a screening approach. Another advantage of a SG screen is that it will report only on candidates with a strong effect on translation since weak alterations of translation will not induce SG formation. Also, we used the SG screen only as a starting point to identify putative regulators of translation but we do not want to claim that these are the only kinases and phosphatases that affect translation rates. We now explained better the rationale for performing a SG screen in our manuscript (line 108-109).

In addition, they identified several kinases and a few phosphatases, but only focused on the CDK1 protein. Did the authors validate any other hits from the screen. As NAGK, HK3, ROS1, and several others have a higher SG score than CDK1, we can assume that they should have a larger effect on translation.

Response: We now included a validation experiment covering 15 of the top candidates, including candidates from all three top categories (**Fig. 2A**). SG induction could be confirmed for four of the candidate KDs (NAGK, ROS1, PNKP and CDK1). We chose to focus on CDK1 primarily because of its central importance for cell cycle progression and mitosis. Moreover, our group has already been working on this kinase based on our discovery that the RNA-binding protein TIAR (a protein known to accumulate in SGs and to control SG dynamics) is essential for the G2/M checkpoint and suppresses CDK1 activity (Lafarga *et al.*, EMBO Reports, 2019, PMID: 30538118).

The authors should address whether the role of CDK1 inhibition on SG formation is directly associated with the alterations in translational control.

Response: We now included quantification of SGs as a time-course experiment following CDK1 inhibition (CDK1i) by Ro3306 treatment (**Fig. 4A**). This experiment shows that SGs form only after prolonged CDK1i (16 and 24 h), whereas translation suppression (quantified over time in Fig. 3C) occurs as early as 1 h after CDK1i. Furthermore, all cell lines we tested so far reduce their translation rate in response to inhibition or KD of CDK1. In contrast, not all cell lines form SGs in response to CDK1i, and we know that SG formation upon CDK1i depends critically on the expression level of one eIF2 α kinase. Since we wanted to focus on the effect of CDK1 on translation in this manuscript, we did not pursue SG formation and its possible link to the cell cycle arrest observed after prolonged CDK1i.

After a siRNA-based screen in HeLa cells, the authors performed all of the mechanistic experiments in the presence of a CDK1 inhibitor. It is unclear why the authors do not use a siRNA-based method to investigate the role of CDK1 in translation. For example, some of the key experiments, especially the polysome profiling and puromycin incorporation, should be repeated in HeLa cells expressing CDK1 siRNA/shRNA.

Response: We now added our results showing that KD of CDK1 by two different siRNAs causes reduced translation as assessed both by polysome profiling and puromycin incorporation (**Fig. 2B–D**). Notably, KD of CDK2, the closest paralog of CDK1, did not lead to reduced translation (Fig. 2B).

For the remainder of our manuscript, we decided to rely on pharmacological inhibition of CDK1 since this allowed us to separate early effects of CDK1i from the late response, which is likely to entail many indirect effects. Translation inhibition is one of the earliest responses to CDK1 inhibition and sets in immediately, while only long-term inhibition induces SG formation, cell cycle arrest and – at some point – cell death. In contrast to the KD and inducible KO of CDK1, which always affects cell viability, short-term inhibition of CDK1 does not affect viability and is fully reversible (see also Vassilev, *Cell Cycle*, 2006, PMID: 17172841).

The experiments delineated in Fig. 2E are not very well controlled. Instead of using 2 different cell lines, it would be better to compare one cell line expressing different CDK1 levels.

Response: Actually, we do not use different cell lines in Fig. 2E (now Fig. 3F). HT2-19 cells were derived from HT1080 cells by deletion of one CDK1 allele and placing the other allele under control of an IPTG-dependent promoter (Itzhaki *et al.*, *Nature Genetics*, 1997, PMID: 9054937). To clarify this, we now included the designation "parental" in Fig. 3F.

Also, the authors should assess whether the CDK1 activity on global protein synthesis is restricted to cancer cell lines or if it is also important in normal cells.

Response: We now included new data on CDK1 inhibition in primary MEFs (**Fig. 3E**), and found that CDK1i causes translation repression as in all other cell models we tested. This includes RPE1 cells (Fig. 4D), MEFs (Fig. S2A, S2C, S2F) and HEK293T cells (Fig. S5E), which are all of non-cancer origin.

In Fig. 4, the authors show that CDK1 inhibitor promotes eIF2 α phosphorylation. For this experiment, it is crucial to recapitulate the increase of p-eIF2 α employing CDK1 KD cells. Is phosphorylation of eIF2 α observed in the phosphoproteomic data.

Response: We analyzed signaling events upon CDK1 KD and found that the results are less pronounced than with acute pharmacological inhibition of CDK1, and more variable. This might be due, as stated above, to the harmful effects of CDK1 KD on cell viability. Since CDK1 KD reports on the long-term consequences of CDK1 inhibition, indirect effects and negative feedback mechanisms are likely to influence signaling events, reduce their magnitude, and make interpretation more difficult. In **Reviewer Fig. R1** (below), we provide an example where eIF2 α phosphorylation is visible upon CDK1 KD. Given our concerns with interpreting these results, we prefer not to include them in our manuscript.

In our phosphoproteomics data, phosphorylation of eIF2 α was not observed. Actually, we neither detected phosphopeptides nor peptides for eIF2 α . Moreover, eIF2 α phosphorylation is strongly visible

only late after CDK1i (16 h and beyond, see Fig. 5A and 5B), whereas phosphoproteomics was performed after 4 h of CDK1i. Therefore, we did not necessarily expect to detect P-eIF2 α in our phosphoproteomics analysis.

Based on the phosphoproteomic analysis, CDK1 should phosphorylate the LARP1 protein. Can the authors validate this result by using western blot analysis?

Response: Unfortunately, there are no antibodies available for the different phosphosites we detected by phosphoproteomics. We tried to detect changes in the overall phosphorylation status of LARP1 using Phostag gels, but since the protein is relatively large (short isoform 1019 amino acids, long isoform 1096 amino acids), we did not succeed in detecting slower migrating bands for LARP1.

In a separate set of experiments, we performed *in vitro* kinase assays using recombinant CDK1/Cyclin B1 and immunoprecipitated Myc-Flag-tagged LARP1. By this approach, we were able to detect phosphorylation of LARP1 by CDK1 *in vitro* (Reviewer Fig. R2, below). By contrast, Flag-tagged PABP did not give a signal in the autoradiogram. Since this is a preliminary result, and phosphorylation of LARP1 by CDK1 requires a more extensive analysis including *in vivo* validation with site-specific antibodies, we prefer not to include this data in the manuscript.

If so, instead of using LARP1^{-/-} HEK293T cells, the authors should use a phospho-deficient mutant of LARP1 in the presence, or absence, of Ro3306

Response: LARP1 has two different isoforms, a short one (isoform 2) that is used in most published studies, and a long one (isoform 1). We first tried to rescue LARP1^{-/-} cells using the short isoform as recently published by Philippe *et al.* (NAR, 2018, PMID: 29244122). However, the response of 5'TOP

mRNAs to CDK1i was not restored in stable LARP1^{-/-} cells re-expressing the short LARP1 isoform. In fact, if one looks at the results from Philippe *et al.* carefully, they observed a shift of 5'TOP mRNAs to lighter fraction in isoform 2 expressing cells only upon Torin treatment, but not under basal conditions. Therefore, we first had to examine if re-expressing the isoform 1 in LARP1^{-/-} cells would fully restore translation control of 5'TOP mRNAs, and we were happy to see that this is the case, both under basal conditions and upon treatment with Ro3306 (new results in **Fig. 8F, S5C and S5D**).

Hence, we are only now in the position to test the regulatory capacity of phospho-deficient and phospho-mimetic mutants of LARP1. Our phosphoproteomics analysis revealed 21 sites in LARP1 whose phosphorylation changes upon CDK1i. We will now generate the corresponding mutants and explore if we can identify specific sites responsible for CDK1-dependent regulation of 5'TOP mRNA translation. This work will require extensive mutagenesis, establishment of numerous LARP1^{-/-} single cell clones re-expressing LARP1 mutants, and careful analysis of 5'TOP mRNA translation by Ribo-Seq or qPCR upon polysome fractionation. These experiments are time consuming and well beyond the scope of this manuscript revision.

Minor Concerns:

The association of CDK1 with the ribosomes was tested by checking the presence of RPS10 in polysomal fractions. What about RPL12?

Response: We now included a new polysome fractionation experiment using RNase I in **Fig. 7A** showing the distribution of CDK1 and RPS3, as well as an alternative experiment using RNase A+T1 in **Fig. S3A** showing the distribution of CDK1, RPS6 and RPL12 within polysome fractions. The shift upon RNase treatment is clearly visible with all ribosomal proteins, and the small proportion of CDK1 that co-migrates with polysomes also shifts to lighter fractions.

The quality of the figures was very poor. Please provide figures with high resolution

Response: We are sorry to learn that the resolution of the figures was not optimal. The resolution of our jpg files was set to 300 dpi, and in our hands, the PDF looked fine. The problem might have come from embedding the jpg files within one Word document. We will now provide single jpg files at 300 dpi and hope that this solves the issue.

Reviewer #2:

The initial purpose of this study was to identify novel regulators of protein synthesis, by siRNA screen against 711 and 256 human kinases and phosphatases, respectively, using HeLa cells stably expressing GFP-G3BP1 which allow to use stress granule formation as a visual read-out. This elegant screen revealed an interesting list of candidates and the authors decided to further pursue CDK1. The study mainly claims that CDK1 enhances global protein synthesis in a cell cycle phase-independent manner to adjust protein synthesis to overall proliferation rate, rather than to a specific phase of the cell cycle. CDK1 was shown before to control protein synthesis mostly during mitosis, but was also suggested to serve as a more global regulator of protein synthesis due to its ability to phosphorylate 4E-BP1 (Heesom *et al.*, 2001; Shuda *et al.*, 2015). Therefore, the claim that CDK1 is a regulator of global synthesis outside of its role as a cell cycle regulator is not fully novel. However, the current manuscript aims to support this claim with by additional strong experiments but the problem is that unfortunately some of these experiments are not fully convincing as presented (see specific comments to Figure 3).

The claim that CDK1 adjusts the overall proliferation rate of the cell to the overall protein synthesis rate is important and interesting. However, first, it is not fully supported by the experiments presented in the current manuscript; second, a recent study by Miettinen T. *et al.*, eLife 2019 titled "Mammalian cell growth dynamics in mitosis" shows that CDK1 leads to increased phosphorylation of 4E-BP1 and cap-dependent protein synthesis during a specific time window in mitosis and convincingly demonstrates the coupling of CDK1-dependent translation regulation to cell proliferation. This

important publication, which was probably published after Haneke et al had submitted their manuscript, must be cited and appropriately discussed.

Response: We are thankful for mentioning the recent study by Miettinen *et al.* (eLife, 2019, PMID: 31063131), which we had not noticed before. In the meantime, also Sun *et al.* (J Biol Chem, 2019, PMID: 31201269) published a study on the role of CDK1-dependent 4EBP1 phosphorylation in translation. The two studies come to different results since Miettinen *et al.* propose that phosphorylation of 4EBP1 during mitosis is responsible for the CDK1-dependent increase in mitotic translation rates, whereas Sun *et al.* propose that the mitosis-specific phosphorylation of 4EBP1 at S83 does not affect cap-dependent translation in mitosis.

To address this issue directly, we now included an analysis of 4EBP1/2 double-KO (DKO) MEFs that were originally generated in the Sonenberg lab (Le Bacquer *et al.*, J Clin Invest, 2007, PMID: 17273556). Our analysis showed that CDK1i led to pronounced translation repression in both WT and 4EBP1/2 DKO MEFs (**Fig. S2C**), and the response was only marginally weaker in the DKO MEFs (**Fig. 6A and 6B**). From this results we concluded that 4EBP phosphorylation may contribute weakly to CDK1-dependent control of global translation, yet it is clear that 4EBP1/2 are not the main effector proteins downstream of CDK1.

In addition, we carried out a new set of synchronisation experiments to quantify the effect of CDK1i on translation in different phases of the cell cycle (**Fig. 4F–H**). As a minimally perturbed system, we synchronised HeLa cells using a double thymidine block, released them from the block for different periods of time, and measured translation (**Fig. 4F**) upon short-term (2 h) inhibition of CDK1 during S-, G2/M- or G1-phase (**Fig. 4G and 4H**). Importantly, these cells were not arrested in the cell cycle but continued proliferating, and the translation rate in mock-treated cells was not reduced compared to asynchronously proliferating cells. Our synchronisation experiments revealed that CDK1 inhibition reduces translation to a similar degree in S-, G2/M- and G1-phase cells, confirming our notion that regulating translation is an extra-mitotic function of CDK1.

We now included the two publications by Miettinen *et al.* and Sun *et al.* in our discussion, and related them to our results with the 4EBP1/2 DKO MEFs (line 383-397). We also made sure to indicate that we are not the first to observe an effect of CDK1 on global translation, rather emphasizing our finding that CDK1 controls translation also outside of mitosis (line 359-370).

The important question the study addresses is how CDK1 elicits its job as a global regulator of protein synthesis. It is stated that "Cdk1 most likely does not act via mTOR" (page 11, line 325-6) which may indeed be the case but this is not convincingly supported experimentally. The authors favor the idea that CDK1 interacts directly with ribosomes, and directly phosphorylates additional ribosomes-associated proteins, including LARP1; but a strong evidence for this notion is lacking (see comments to Figure 5).

Response: We agree with the reviewer that the manuscript would benefit from additional data clarifying the role of mTOR signaling downstream of CDK1. We now provide several lines of evidence supporting our notion that CDK1 regulates translation independently of mTOR:

First, CDK1 inhibition does not affect the interaction between eIF4E and eIF4G, in contrast to mTOR inhibition. This result was shown in the supplement of our original manuscript; we now moved it to the main part (Fig. 5F–H).

Second, we now added new results obtained in MEFs expressing a hyper-active mTOR C1483F mutant (**Fig. 6C**). Elevated S6K1 phosphorylation levels confirmed enhanced activity of the mTOR C1483F mutant in presence of the CDK1 inhibitor Ro3306 (**Fig. 6D**). However, translation repression upon CDKi was not affected by expression of the mTOR C1483F mutant (**Fig. 6E and S2D**), demonstrating that CDK1 controls global translation independently of mTOR. This observation is in perfect agreement with our finding (described above) that 4EBP1/2 are not major effectors controlling translation downstream of CDK1 (Fig. 6A, 6B and S2C).

Third, CDK1 acts differently on LARP1 than mTOR. Our new LARP1 RNA-IP results show that Ro3306 treatment does not increase the binding of 5'TOP mRNAs to LARP1 (**Fig. S5F**). In contrast, mTOR inhibition was shown to enhance binding of 5'TOP mRNAs to LARP1 (Philippe *et al.*, NAR, 2018, PMID: 29244122). Also, the Ro3306-sensitive phosphorylation sites we identified in LARP1 are different from the described mTOR sites. The major sites we identified include minimal CDK

consensus sites (S*/T*P, S*/T*PXR/K) but do not match the mTOR consensus (R/KXR/KXXS*/T*). Taken together, our results now provide much stronger evidence that CDK1 controls translation independently of mTOR, which contrasts models that are currently discussed in the literature.

The strongest section in the manuscript is the Ribo-Seq analysis and related experiments (Figs 6, S6, S7), which uncovered that CDK1 exerts a particularly strong effect on the translation of 5'TOP mRNAs. The effect of CDK1 on 5'TOP mRNA was convincingly demonstrated to be LARP1-dependent. The suggestion is that LARP1 phosphorylation itself is dependent on CDK1 activity, but this specific point remains open.

Response: Our phosphoproteomics analysis clearly showed several CDK1-dependent phosphorylation sites on LARP1 (experiment n1 in Fig. S3B). Unfortunately, there are no antibodies available for the different phosphosites we detected, and hence we could not easily validate this result. We tried to detect changes in the overall phosphorylation status of LARP1 using Phostag gels, but since the protein is relatively large (short isoform 1019 amino acids, long isoform 1096 amino acids), we did not succeed in detecting slower migrating bands for LARP1.

In a separate set of experiments, we performed *in vitro* kinase assays using recombinant CDK1/Cyclin B1 and immunoprecipitated Myc-Flag-tagged LARP1. By this approach, we were able to detect phosphorylation of LARP1 by CDK1 *in vitro* (see **Reviewer Fig. R2** above in the response to reviewer #1). By contrast, Flag-tagged PABP did not give a signal in the autoradiogram. This result suggests that LARP1 might be a direct target of CDK1, yet validation experiments using site-specific antibodies will be required to confirm this *in vivo*. Given that the *in vitro* phosphorylation of LARP1 by CDK1 is a preliminary result, we prefer not to include this data in the manuscript.

However, we were very pleased to achieve a rescue of CDK1-dependent 5'TOP mRNA regulation by expressing the long form (isoform 1) of LARP1 in LARP1^{-/-} cells (new results in **Fig. 8F, S5C and S5D**). In our hands, the short isoform (isoform 2), which is used in most publications, did not restore CDK1-dependent control of 5'TOP mRNA translation. Interestingly, isoform 2 was shown to rescue mTOR-dependent control of 5'TOP mRNA translation in LARP1^{-/-} cells (Philippe *et al.*, NAR, 2018, PMID: 29244122). This difference is another indication that CDK1 and mTOR regulate LARP1 through distinct mechanisms. Since our phosphoproteomics analysis revealed 21 sites in LARP1 whose phosphorylation changes upon CDK1i, identifying the relevant sites and their importance for regulation of LARP1 by CDK1 will require an extensive analysis that is beyond the scope of this manuscript revision.

Overall, this is an important and interesting study with partial novelty, which suffers from major problems as detailed below.

Comments:

(*) General comment about Statistics:

When the same control sample(s) are used in an experiments addressing more than one variable (condition), there is a requirement to either use Anova test (for example for Figs 2D, 4B,D) or to use student t-test with Benferroni correction. In addition, a paired Student's t-test should be used (unless there is a justified excuse to use unpaired t-test. Given the above, the statistics of many experiments presented in the manuscript must be re-calculated followed by changing some of the conclusions. For example, correct analysis of Fig. 4 reveals that phosphorylation level of eIF2a upon 4h treatment with Ro33606, is not significant (P=0.08).

Response: Whenever possible we used a paired instead of an unpaired t-test. Sometimes, however, not all conditions (e.g. not all time points or not all siRNAs) were tested in every repeat. In these cases, we have to compare unequal repeat numbers with each other, which can only be done by unpaired testing. In all cases where every condition has equal repeats, a paired t-test was performed.

For the unpaired t-tests, we now chose unequal variance (Welch's t-test), which is the conservative option in cases where it is unclear whether the variance between samples is equal or not. The test chosen for each analysis is specified in the figure legends.

In Fig. 3B, 5B, 5E and 5H, we normalized all values to the 0 hour time point or DMSO control. Thereby, we lost the standard deviation for the control samples and thus performed a one-sample t-

test, which is correct. However, we noticed that we have to log-transform all values prior to testing, because ratios are expected to follow a log-normal and not a normal distribution. This version of the one-sample t-test is identical to a ratio paired t-test, and all p-values in Fig. 3B, 5B, 5E and 5H were now calculated according to the ratio paired t-test.

As suggested by the reviewer, we also analyzed these time course experiments, after log-transformation of all values, using one-way ANOVA and Bonferroni's multiple comparisons test (comparing each value to the 0h, DMSO-treated control value, which was set to one). The p-values are shown below in comparison to the p-values of the ratio paired t-test:

Fig. 3B	One-Way ANOVA	Bonferroni's multiple comparisons test	ratio paired t-test, one-tailed
all vs. all	0.0514		
0 vs. 0.5h		>0.9999	0.0687
0 vs. 1h		0.8772	0.0708
0 vs. 4h		0.3062	0.0577
0 vs. 16h		0.0158	0.0533

Fig. 5B	One-Way ANOVA	Bonferroni's multiple comparisons test	ratio paired t-test, one-tailed
all vs. all	0.0007		
0 vs. 1h		0.1836	0.0082
0 vs. 4h		0.1260	0.0312
0 vs. 8h		0.0955	0.1097
0 vs. 16h		0.0029	0.0125
0 vs. 20h		0.0026	0.0076
0 vs. 24h		0.0001	0.0014
0 vs. 28h		0.0016	0.0010

Fig. 5E 4EBP1	One-Way ANOVA	Bonferroni's multiple comparisons test	ratio paired t-test, one-tailed
all vs. all	0.0817		
0 vs. 1h		0.4882	0.0471
0 vs. 4h		0.0867	0.0102
0 vs. 16h		0.0745	0.0521
Fig. 5E RPS6	One-Way ANOVA	Bonferroni's multiple comparisons test	ratio paired t-test, one-tailed
all vs. all	0.0002		
0 vs. 1h		0.1339	0.0007
0 vs. 4h		0.0028	0.0058
0 vs. 16h		<0.0001	0.0053

Fig. 5H	One-Way ANOVA	Bonferroni's multiple comparisons test	ratio paired t-test, one-tailed
all vs. all	0.0023		
DMSO vs. R4		>0.9999	0.1156
DMSO vs. R16		>0.9999	0.2475
DMSO vs. T4		0.0016	0.0086
all vs. all	0.0601		

DMSO vs. R4		>0.9999	0.0702
DMSO vs. R16		>0.9999	0.3770
DMSO vs. T4		0.0507	0.0426
all vs. all	0.7029		
DMSO vs. R4		>0.9999	0.4199
DMSO vs. R16		>0.9999	0.3332
DMSO vs. T4		0.8732	0.1960

Comparing the p-values from the ratio paired t-test and ANOVA, we see that several p-values are larger by ANOVA-Bonferroni, yet there are also may cases where the p-values are smaller. In our experience, translation is quite sensitive to subtle differences between repeat experiments (e.g. cells tend to grow better when left alone in the incubator over the weekend, cells react to the addition of fresh medium, to the quality of serum batches etc...). Therefore, pairwise testing of our measurements is important, and since ANOVA does not take into account the pairwise character of our data, we think that a ratio paired t-test is more suitable.

To address the question of appropriate testing, we suggest to follow the advice by Harvey Motulsky, *Intuitive Biostatistics*, Third Edition, page 185. Here, Motulsky explains that *corrections for multiple comparisons are not essential if only a few planned comparisons are performed. The danger lies in testing all or many possible combinations of conditions and then reporting only the significant differences.* Since we typically compare only two conditions in our experiments (e.g. DMSO vs. Ro3306 treatment, control siRNA vs. CDK1 siRNA, control cell line vs. KO cell line), multiple comparison testing is not necessary. Moreover, we now show all p-values in every figure, irrespective of any (arbitrary) significance level, so that the reader can judge him/herself which results are convincing. Also, we did not test all possible comparisons and then reported only on the significant ones. Rather, we determined beforehand which comparisons are interesting, and then tested these directly. Given that we focused on very few comparisons, correction for multiple comparisons is not necessary.

In the new Fig. 2C, we also applied the ratio paired t-test because these values are also normalized to the S14 control siRNA condition.

In the experiments where we compared fold changes with each other (fold change in polysomal ribosomes upon R03306 treatment between cell lines; Fig. 5C, 6A, 6E, 6G, 6H, S5E), we now log-transformed the ratios to achieve normal distribution, and then applied a paired Student's t-test. This allows us to directly compare the fold changes between different cell lines, which is the main goal of these experiments.

As suggested by the reviewer, we also performed two-way ANOVA on the original polysome profile measurements (Fig. S2A, S2C, S2D, S2E, S2F, S5E); the three p-values from ANOVA are shown below in comparison to the p-values of the paired t-test:

	Paired t-test of log(fold-changes)	Two-way ANOVA		
		interaction	cell line effect	treatment effect
eIF2 α (SS vs. AA)	<0.001	0.1336	0.1967	<0.0001
4EBP1/2 (WT vs KO)	0.11	0.0596	0.259	<0.0001
Parental vs. mTOR WT	0.20	0.7329	0.4135	0.0089
Parental vs. mTOR C1483F	0.11	0.542	0.6405	0.0014
Parental vs. S6K1 WT	0.003	0.0237	0.3379	<0.0001
Parental vs. S6K1 CA	0.002	0.0216	0.2494	<0.0001
RPS6 MEFs (P ^{+/+} vs. P ^{-/-})	0.34	0.7717	0.7256	0.0003
LARP1 (WT vs. KO)	0.006	0.552	0.2575	0.0002

When comparing the p-values of the paired t-test with the interaction p-values from ANOVA (which tests whether the effect of Ro3306 treatment is different between cell lines), we notice that the paired t-test tends to give lower p-values, although not in every case. Again, we reasoned that pairwise testing is important in our experimental setting, and therefore decided to use the paired t-test of the log-transformed fold-changes).

General comment about graphs presentation

Instead of showing only the fold effects, it is important to present the entire data, including that of the untreated (or DMSO-treated) cells.

Response: We show fold-changes in the main figures since this is the easiest way to compare the effect of Ro3306 on translation between cell lines (Fig. 6A, 6E, 6G, 6H, S5E). The entire datasets are presented in the supplementary figures (Fig. S2A, S2C, S2D, S2E, S2F, S5E), which also contain the values for the control conditions. Moving the entire datasets to the main figures would have made them very bulky.

Specific comments

Figure 3

- % SG in untreated cells is not shown (Fig 3A). It might be 10% even without Ro3306 treatment.
- Fig 3A shows one field for each cell type. It is not clear if the cells were synchronized for this experiment or not. In the left image most of the cells are in G1 and one cell contains SG. In the right image most of the cells are in S/G2/M, while >3 cells contain SG, some of them seem to be at late stages of mitosis.

Response: This comment now pertains to Fig. 4B and 4C. We added the entire dataset of this experiment in **Fig. S1C**, which shows that DMSO-treated HeLa cells do not form SGs and that Ro3306 treatment for 16 hours, as expected, induces a strong cell cycle arrest at the G2/M boundary. Since it is rather complicated to extract the essential information on SG frequency in Cdt1- and Geminin-positive cells from the entire dataset in Fig. S1C, we kept the simpler representation of the normalized data in Fig. 4C.

In addition, we also included new FACS data on the effect of Ro3306 treatment on the cell cycle in HeLa cells (Fig S1B). The analysis shows little changes in cell cycle distribution after 1 or 4 hours, but a prominent cell cycle arrest after 16 hours of Ro3306 treatment.

- Cells at early mitosis (just upon nuclear membrane break down) must be presented in the chosen field. They are expected to express mVenus-Geminin throughout the cell, which will combine with the color of eIF3B detector. It is important to present these cells in order to address the question related to SG status during mitosis

Response: CDK1 inhibition hinders cells from entering into mitosis. Therefore, we hardly see cells in early mitosis upon 16 hours of Ro3306 treatment. Furthermore, mitotic cells do not make SGs as has been well documented by Sivan *et al.* (Mol Cell Biol, 2007, PMID: 17664278).

- The statistics of this experiment is not clear (Fig. 3B: N=? and see also general comment about statistics *)

Response: For Fig. 3B (now Fig. 4C), we calculated the % of SG containing cells of all marker-positive cells. Four independent repeats were performed and statistical significance was determined by paired, two-tailed Student's t-test. This information is given in the legend of Fig. 4C.

- The text at the bottom of page 6 does not match Fig. 3C and its legend. There is a confusion between arrest at G1 or S phase, and only 2 out of the 3 profiles are shown.

Response: This comment now pertains to Fig. 4E. We thank the reviewer for pointing out this contradiction. By double thymidine block, cells are arrested at the G1/S boundary. We rephrased the statement in the figure legend (line 542 and 367) and in the text (line 192). Only 2 profiles are shown, this is correct. In the text (line 194), we additionally refer to the profile of unsynchronized cells, shown in Fig. 3C.

Figure 4

- Fig 4B-C-D-E: see the general comment related to statistics (*)

Response: Fig. 4B–D is now Fig. 5B, 5E, 5C, 6G and 6H. In Fig. 5B and 5E, all values were normalized to the control value (0 hour time point), and a ratio paired t-test was performed on the log-transformed values. For the statistics in Fig. 5C, 6G and 6H, the fold-changes were log-transformed and subjected to a paired Student's t-test. In both cases, we consider the t-test more appropriate than ANOVA because it takes into account the pairwise character of the data. Please also refer to our response to the general comment on statistics (above).

- Fig 4A: Quantification of some of the bands is problematic: (i) some are fused between lanes. It may be due to over loading and/or overexposure. Is the quantification within the linear range? (ii) it is difficult to understand which of the different P-4E-BP1 bands represent the phosphorylated forms and it is difficult to separate them for quantification; (iii) total eIF2 α blot suffer from a strong background.
- The effect of Ro3306 after 24h is presented in Fig4A and mentioned in the main text, while statistical calculations are missing for this time point in Fig. 4B.
- 4E-BP1 phosphorylation decreases upon CDKi, not increases as indicated in page 7 line 186.

Response: This comment now pertains to Fig. 5A, 5B, 5D, 5E. We repeated these Western blots and provide new examples in Fig. 5A and 5D. We tried to better distinguish the different P-4EBP1 bands, yet using higher percentage gels did not improve the resolution. Sun *et al.* (J Biol Chem, 2019, PMID: 31201269) carefully addressed the different P-4EBP1 isoforms in mitosis and their dependency on CDK1 activity, using 2D gel electrophoresis to distinguish the different isoforms. In our manuscript, we do not want to focus on how CDK1 regulates 4EBP1 phosphorylation; this has already been analyzed in detail by many groups. We do note, however, that prolonged treatment with Ro3306 causes an increase in the P-4EBP1 signal intensity (quantified in Fig. 5E), while the protein accumulates in one of its lower migrating (hypophosphorylated) isoforms. The simplest explanation is that a single 4EBP1 site becomes strongly phosphorylated, while the other sites lose their phosphorylation. To make sure that our statement is not misleading, we re-phrased the description of 4EBP1 phosphorylation on line 224-226 as follows: "4EBP1, a direct target of mTORC1, **showed a change in the phosphorylation pattern** upon CDKi, and accumulated in a hypophosphorylated form 16 h **after** Ro3306 treatment (Fig. 5D and 5E). Since 4EBP1/2 double knockout had only a minor effect on translation repression by CDKi (new result in Fig. 6A), we did not pursue the role of 4EBPs in control of global translation by CDK1.

In addition, we now provide quantification for all time points in the Western blots (Fig. 5B, 5E), and we included a new WB example for P-eIF2 α and total eIF2 α that does not suffer from strong background signal in Fig. 5A.

Figure 5

Fig 5A: the data is not convincing and very confusing.

- It is important to show the polysomal fractions that were used in the main Figure rather than in a Supp Fig.
- Although RNase treatment is expected to cause polysomes disassembly, polysomes are still mostly stable, based on the presence of RPS10 in fractions 7-8-9-10.

Response: We now show two different examples for the association of CDK1 with polysomes, using two different ways to disassemble polysomes (RNase I in Fig. 7A, RNase A/T1 in Fig. S3A). We also aligned the polysome profiles to the blots and included them in the figures. In our experience, it is difficult to achieve complete disassembly of polysomes by RNase treatment.

Fractions of each of the untreated or RNase treated experiments, should be analyzed for RPS10 and CDK1 using different positions of the same lanes (following SDS-PAGE and blotting). However, according to the different pattern of the bands on the blots, it looks like RPS10 and CDK1 were detected using different gels.

Response: In every experiment, the same blot was used for probing CDK1 and the different ribosomal proteins. The small ribosomal proteins run at the very bottom of the gel and therefore show a different pattern of bands than CDK1, which runs in the middle of the gel. CDK1 and RPS6 are similar in size (34 kDa and 26 kDa, respectively). We therefore included RPS6 in addition to RPL12 in Fig. S3A.

Why there is no RPS10 signal specifically in fraction 2 in both untreated and RNase treated experiments? Is there a biological explanation?

Response: Fraction one and two of our gradients contain proteins and complexes with a density below that of 40S. Presumably, RPS10 migrating in the first fraction corresponds to the free protein, whereas RPS10 incorporated into 40S migrates in fractions 3–12. Since this distribution is a bit unusual for ribosomal proteins, we omitted the RPL10 blot and instead used RPS3 (Fig. 7A), RPS6 and RPS12 (Fig. S3A), none of which are present as free proteins in the first fraction.

Page 8, line 214-217: The strength of the statement related to association of CDK1 with polysomes does not fit the strength of the data shown in Fig5A.

Response: We now removed the sentence "First, we explored if CDK1 indeed interacts with ribosomes" and on line 266-268 now only state that "Polysome profile analysis revealed that a small proportion of CDK1 co-migrates with polysomes, and shifts to lighter fractions upon disassembly of polysomes by RNase treatment (Fig. 7A and S3A)". We further phrased the statement on this observation in the discussion on line 403-405 more carefully, as follows "Interestingly, we observed that a small proportion of CDK1 co-sediments with polysomes (Fig. 7A), and association of CDK1 with ribosomes was also reported by a mass spectrometry approach (Simsek et al., 2017)".

Fig 5B:

- It is not mentioned how many replicates were used for MS detection. Moreover, there is no information about MS data analysis (FDR %, normalization, statistics, etc.). Therefore it is impossible to judge the strength of the statement in page 8, line 227 "A prominent reduction of phosphorylation upon CDKi was observed for RPS6 and LARP1". Some of the RPS6 and LRRP1 phosphopeptides are located around the zero number of the 5B graph while an explanatory comment about the meaning of this observation is absent. Moreover, the authors do not comment about RPS6 protein abundance which seems to slightly increase in response to CDK1i. In case this is due to statistics considerations (which are not mentioned), maybe some of the changes in phosphorpeptides abundance should be ignored too, based on similar statistics considerations.
- Fig. S5B should be part of the main figure and an additional test should be added to explain the differences in location of the indicated phosphoproteins on the graphs.

Response: We now combined the two phosphoproteomics experiments (n1 and n2) into a single plot (Fig. 7B), which shows only those phosphopeptides that were detected in both n1 and n2. The individual experiments are now shown in Fig. S3B). Statistics was performed within n1 and n2 as light and heavy peptides were measured 3 times in each experiment. Statistics across the two experiments would have been very problematic since the sensitivity was vastly different in n1 (2918 phosphopeptides detected) and n2 (273 phosphopeptides detected). In the text, we took out the word "prominent" and now state " Both analyses revealed a reduction in phosphorylation of RPS6, LARP1, DAP and Lamin A in response to CDK1i (Fig. 7B)" (line 275-276).

The abundance of RPS6 was not affected by Ro3306 treatment in experiment n2, and our Western blot analyses did not show a clear reduction of RPS6 levels upon Ro3306 treatment.

Minor comments:

- Page 6 lines 136-138: To enhance clarity the following sentence should be changed to: "CDK1 levels were reduced at least 2-fold in HT2-19 cells cultured in presence of IPTG compared to the parental HT-1080 cells, and polysomal ribosomes decreased from 43% to 32%"

Response: We changed the sentence according to the reviewer's suggestion; indeed the statement is now easier to understand (line 158-160).

- Fig 2BCD: multiple treatment times with Ro3306 not indicated in the legend

Response: This now refers to Fig. 3B and 3C. Since the time of Ro3306 treatment is indicated within each figure, we decided not to repeat this information in the figure legends. Given that our manuscript is already longer than requested by the guidelines of JCB, we need to be restrictive with adding text.

- Fig S1: Add to the legend what the arrows indicate (also add the appropriate information to the text in page 6 line 140)

Response: This comment now refers to Fig. S1A. The arrows indicate the increase in cell size. This information was added to the legend of Fig. S1A.

- Fig S2: Add to the legend what the arrows indicate

Response: This comment now refers to Fig. S1D. The arrows point towards the cilia. This information was added to the legend of Fig. S1D.

- Fig S3A: The graph is confusing. SS and AA cells should be presented in two separated graphs because they are not compared to each other (as Anova test was not performed). Note that this comment also applies to additional Figures throughout the manuscript.

Response: We replied to the suggested use of ANOVA in response to the general comment related to statistics (see above). Also, we would like to point out that the polysome profiles of different cell lines were combined in a single graph in order to save space. We believe that the color code allows following the curves of several cell lines and conditions in one graph.

Summary of major changes in the revised manuscript JCB #201906147

1. We added our validation results of the stress granule (SG) screen (Fig. 2A) in order to better explain why we chose to focus on the role CDK1 in translation control. 15 of the top candidates from the primary screen were selected (including candidates from all three top categories) and SG formation was measured upon knockdown (KD). In these validation experiments, many candidates turned out to be false positives, yet KD of CDK1, NAGK, ROS1 and PNKP reproducibly induced SG formation. We then chose to focus on CDK1 primarily because it is of central importance for cell cycle progression and mitosis, but also because our group is already working on this kinase based on our discovery that the RNA-binding protein TIAR (a protein known to accumulate in SGs and to control SG dynamics) is essential for the G2/M checkpoint and suppresses CDK1 activity (Lafarga et al., EMBO Reports, 2019).
2. We included new data on translation rates in CDK1 KD cells (Fig. 2B–D). Polysome profile analysis upon CDK1 KD showed a strong reduction in the percentage of polysomal ribosomes, whereas control KD or CDK2 KD did not (Fig. 2B). CDK2 was included because it is the CDK family member closest to CDK1. Puromycin incorporation was also reduced upon CDK1 KD in comparison to control KD (Fig. 2C). Together, the new data clearly show that CDK1, but not CDK2, affects global translation rates.
3. We confirmed that CDK1 inhibition also affects translation rates in primary mouse embryonic fibroblasts by using polysome profile analysis, demonstrating that CDK1 enhances global translation also in non-transformed cells (Fig. 3E).

4. We now added representative FACS data of HeLa cells treated with the CDK1 inhibitor Ro3306 for different periods of time. The analysis shows that Ro3306 treatment for 16 hours leads to a strong cell cycle arrest, whereas short-term treatment for up to 4 hours does not cause a visible accumulation of cells at the G2/M boundary (Fig. S1B). Since global translation is reduced already after one hour of Ro3306 treatment (Fig. 1B and 1C), cell cycle arrest per se cannot be responsible for the effect on translation.
5. We added a quantification of SG-containing cells after different periods of Ro3306 treatment (Fig. 4A). We also included the non-normalized data (Fig. S1C) of our SG analysis in FUCCI cells (Fig. 4B and 4C), whereby readers can see the changes in the distribution of Cdt1-positive and Geminin-positive cells due to cell cycle arrest at the G2/M boundary upon long-term (16 h) inhibition of CDK1.
6. We carried out synchronisation experiments to quantify the effect of CDK1 inhibition on translation in different phases of the cell cycle (Fig. 4F–H). As a minimally perturbed system, we synchronised HeLa cells using a double thymidine block, released them from the block for different periods of time, and measured translation (Fig. 4F) upon short-term (2 h) inhibition of CDK1 during S-, G2/M- or G1-phase (Fig. 4G and 4H). Importantly, these cells were not arrested in the cell cycle but continued proliferating, and the translation rate in mock-treated cells was not reduced compared to asynchronously proliferating cells. These synchronisation experiments revealed that CDK1 inhibition reduces translation to a similar degree in S-, G2/M- and G1-phase cells, confirming our notion that regulating translation is an extra-mitotic function of CDK1.
7. We added additional time course experiments in order to improve quantification of the phosphorylation level of eIF2 α (Fig. 5A and B), RPS6 and 4EBP1 (Fig. 5D and 5E). Also, shorter exposure times of Western blots are shown.
8. To test the importance of 4EBPs in regulating translation downstream of CDK1, we included new polysome measurements in 4EBP1/2 double knockout (KO) MEFs (Fig. 6A, 6B, S2C). The result shows that translation repression upon CDK inhibition is barely affected by the absence of 4EBP1/2, indicating that 4EBPs play only a minor role in CDK1-dependent regulation of global translation.
9. To further clarify the role of mTOR as a possible signaling pathway controlling translation downstream of CDK1, we also examined translation in MEFs expressing a hyper-active mTOR C1483F mutant (Fig. 6C, 6D, 6E, S2D). The results demonstrate that CDK1 controls translation independently of mTOR.
10. We exchanged the figure demonstrating that CDK1 associates with polysomes with a new experiment (Fig. 7A), and added an additional experiment where we used RNase A and T1 instead of RNase I, which improved polysome disassembly (Fig. S3A). The results show more clearly that a small portion of CDK1 co-migrates with polysomes and shifts to lighter fractions upon RNase treatment.
11. We now decided to show the results from the two phosphoproteomics experiments in a combined graph (Fig. 7B), the individual experiments are depicted in Fig. S3.
12. We performed a LARP1 rescue experiment by re-introducing a LARP1 cDNA into LARP1-/- HEK293T cells. Polysome fractionation followed by qPCR analysis showed that re-expression of LARP1 restores Ro3306-dependent suppression of 5'TOP mRNA translation (Fig. 8E, 8F and Fig. S5). We also tested whether binding of LARP1 to 5'TOP mRNAs is affected by CDK1 inhibition, yet did not observe a difference (Fig. S5F). In conclusion, the rescue experiment provides strong evidence that LARP1 is downstream of CDK1 in controlling 5'TOP mRNA translation. The detailed mechanism by which CDK1 regulates the activity of LARP1 will require detailed mapping of the phosphorylation sites on LARP1 and functional studies on corresponding LARP1 mutants, which is beyond the scope of this revision.
13. We revised our statistical analyses by log-transforming values that represent ratios or fold changes before performing t-tests. Also, we switched to unequal variance whenever an unpaired t-test was performed. All tests are specified in the figure legends. The altered analysis led only to small changes in the p-values, which are now depicted throughout all figures, and did not affect the main results and conclusions of our study. In the response to reviewer #2, we provide a detailed discussion and justification of our statistical approach.
14. Due to the limited number of supplementary files, we now deposited the Excel tables with the raw data of the SG screen and the phosphoproteomics experiments in a repository of Heidelberg

University (<https://doi.org/10.11588/data/EFHOBZ>). The raw data of the ribosome footprint are deposited in the GEO database (accession number: GSE128538).

December 12, 2019

RE: JCB Manuscript #201906147R

Prof. Georg Stoecklin
Medical Faculty Mannheim of Heidelberg University
Biochemistry
Ludolf-Krehl-Strasse 13-17
CBTM
Mannheim 6968167
Germany

Dear Prof. Stoecklin:

Thank you for submitting your revised manuscript entitled "CDK1 couples proliferation with protein synthesis". We would be happy to publish your paper in JCB pending final revisions necessary to meet our formatting guidelines (see details below).

- Provide main and supplementary text as separate, editable .doc or .docx files
- Provide figures as separate, editable files according to the instructions for authors on JCB's website, paying particular attention to the guidelines for preparing images at sufficient resolution for screening and production

Thank you for your attention to these final processing requirements. Please revise and format the manuscript and upload materials within 7 days or contact us to let us know if you require more time.

A. MANUSCRIPT ORGANIZATION AND FORMATTING:

Full guidelines are available on our Instructions for Authors page, <http://jcb.rupress.org/submission-guidelines#revised>. **Submission of a paper that does not conform to JCB guidelines will delay the acceptance of your manuscript.**

B. FINAL FILES:

- An editable version of the final text (.DOC or .DOCX) is needed for copyediting (no PDFs).
- High-resolution figure and video files: See our detailed guidelines for preparing your production-

ready images, <http://jcb.rupress.org/fig-vid-guidelines>.

Thank you for this interesting contribution, we look forward to publishing your paper in Journal of Cell Biology.

Sincerely,

Ian Macara, Ph.D.
Editor

Marie Anne O'Donnell, Ph.D.
Scientific Editor

Journal of Cell Biology

Reviewer #2 (Comments to the Authors (Required)):

The revised manuscript presents an important function of CDK1 as a connector of global protein synthesis and cell proliferation, independently of mTOR, which does similar job connecting protein synthesis and cell growth. While both connectors share downstream effectors, it appears that their function is independent from each other. Experiments presented in the revised version support the conclusion that CDK1 (but not CDK2) regulate translation globally, i.e., outside of mitosis, and the conclusion that CDK1 function in global translation regulation does not dependent of mTOR activity. Importantly, additional data provides stronger evidence that LARP1 is downstream of CDK1 in controlling 5'TOP mRNA translation, i.e. that CDK1 and mTOR regulate LARP1 by different mechanisms.

New data shows that CDK1 inhibition affects translation rates also in non-transformed cells (primary MEFs). It is also presented more clearly that a small portion of CDK1 co-migrates with polysomes, bringing up initial clues about CDK1-mediated phosphorylation of ribosomal or ribosome-associated proteins.

The authors took great care of statistical approach used for data analysis, which significantly enhances the quality of their conclusions in the revised version.

The response to each and every comment raised by the reviewers is adequate and satisfactory.

Therefore, the revised version is appropriate for publication in JCB without any further changes.